# GENERATIVE CONFORMAL PREDICTION WITH OPTIMIZED COVERAGE ALLOCATION

## ABSTRACT

Conformal prediction provides model-agnostic uncertainty quantification with guaranteed coverage, but conventional methods often yield overly conservative uncertainty sets, particularly in multimodal or heterogeneous settings. This inefficiency arises from two sources: (i) limited expressiveness of the predictive model and (ii) simplistic nonconformity scores design. Most existing approaches advance only one of these axes, leaving the other underexplored. We propose *generative conformal prediction with Optimized Ranking and Coverage Allocation (ORCA)*, a three-stage framework that advances both aspects jointly. ORCA leverages generative models to capture the full conditional distribution and introduces a rank-dependent optimization procedure that adaptively allocates coverage for efficiency while maintaining validity. We cast this coverage allocation as an optimization problem, derive an exact mixed-integer linear programming formulation, and show that the solution converges asymptotically to the oracle density-level set. Across synthetic, semi-synthetic, and real datasets, ORCA produces substantially more efficient uncertainty sets than state-of-the-art baselines, demonstrating robust gains in scenarios where conventional conformal prediction methods fail.

## 1 INTRODUCTION

Conformal prediction (CP) (Vovk et al., 2005) has gained prominence as a powerful framework for uncertainty quantification, offering uncertainty sets with statistically guaranteed coverage under minimal assumptions. As a model-agnostic approach, CP can enhance any black-box model to provide uncertainty sets that contain the ground truth with a user-specified probability. This versatility has driven its adoption across diverse domains, inclduing healthcare (Seoni et al., 2023), finance (Fujimoto et al., 2022; Blasco et al., 2024), and autonomous systems (Michelmore et al., 2020; Su et al., 2023; Grewal et al., 2024), where reliable decision-making under uncertainty is critical.

Despite its strengths in coverage validity and model-agnostic applicability, CP often yields overly conservative uncertainty sets, limiting practicality and interpretability when the conditional distribution $p(Y \mid X)$ exhibits complex structure such as multimodality or heterogeneity. This inefficiency arises from two sources: (i) limited model capacity to capture the geometry of the conditional distribution, and (ii) simplistic nonconformity scores that fail to exploit distributional richness. For instance, linear regression provides only conditional mean estimates, whereas quantile regression could recover multiple conditional quantiles and thus reveals more of the distribution. Likewise, for a fixed regression model, one may choose a simple absolute residual as the score, or a more sophisticated variant such as the locally weighted nonconformity measure that normalizes residuals to improve efficiency (Papadopoulos et al., 2002). Both directions of using more expressive models and designing more refined scores enhance uncertainty quantification. Recent work has pushed along these axes; however, most approaches treat model expressiveness and score design in isolation, advancing one while neglecting the other. In this work, we unify these perspectives by proposing a framework that couples generative modeling with rank-dependent, vectorized score constructions optimized for efficient coverage allocation.

In this paper, we propose *generative conformal prediction with Optimized Ranking and Coverage Allocation (ORCA)*, a three-stage framework that combines generative modeling with optimization-based and efficiency-oriented non-conformity score design. ORCA leverages generative samples to capture the full conditional distribution and introduces a rank-dependent optimization procedure that

adaptively allocates coverage for greater efficiency while maintaining exact finite-sample validity. This yields prediction sets that fully adapt to the geometry of $Y \mid X$, thus leading to superior efficiency. Figure 1 highlights our advancements over classical conformal prediction baselines, illustrating how ORCA achieves a more adaptive and efficient uncertainty set.

In summary, the main contributions of this paper are:

- **Generative conformal prediction.** ORCA employs generative models as predictors to capture distributional complexity beyond mean- or quantile-based approaches. By leveraging generative sampling with local density proxy ranking mechanism, ORCA exploits the fitted conditional distribution, encoding finer distributional structure into the conformal procedure.
- **Optimized coverage allocation.** Rather than collapsing generated sample residuals into a single statistic, ORCA operates on rank-dependent residual vectors. We cast conformal coverage as an explicit coverage allocation optimization problem and reformulate it as a mixed-integer linear program (MILP), which ensures global optimality with practical scalability.
- **Theory with oracle guarantees.** We prove that ORCA maintains exact finite-sample validity and, asymptotically, converges to the oracle optimal highest-density region, thereby establishing a rigorous link between optimization-based conformal prediction and density-level sets.
- **Empirical evaluation.** Experiments on synthetic and real datasets show that ORCA achieves much more efficient uncertainty sets than baselines while maintaining exact coverage.

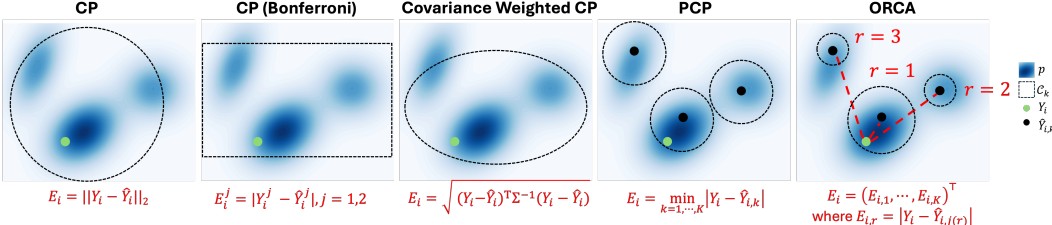

Figure 1: Comparison of conformal prediction methods and their prediction regions. From left to right: standard CP (Vovk et al., 2005), Bonferroni-corrected CP (Neeven & Smirnov, 2018), Covariance Weighted CP (Messoudi et al., 2022; Xu et al., 2024), PCP (Wang et al., 2023), and ORCA (our method). The background gradient shows the multimodal data distribution. ORCA adjusts circle radii via ranking—larger in dense areas, smaller in sparse ones—yielding more efficient and flexible prediction sets. Green dots denote true responses ($Y_i$), black dots are generated samples ($\hat{Y}_{i,k}$), and dashed lines mark prediction boundaries ($\mathcal{C}_k$).

**Related Work**  Extended works have been done focusing on CP for deterministic models. Different summary statistics of the conditional distribution have been studied, for example, mean (Shafer & Vovk, 2008; Lei et al., 2018) and quantiles (Romano et al., 2019; Kivaranovic et al., 2020; Alaa et al., 2023). Sesia & Romano (2021) estimate conditional distributions with histograms to obtain efficient intervals with approximate conditional coverage. Though the above methods all provide continuous prediction intervals, they cannot adapt to multimodality well. In Lei et al. (2013) and Lei & Wasserman (2014), the authors first consider the aspect of density level sets and density-based clustering for providing efficient prediction set. The authors use kernel density estimator to construct the non-conformity scores and with a proper choice of bandwidth, the prediction set could be discontinuous. There are also following works (Izbicki et al., 2020; Chernozhukov et al., 2021; Izbicki et al., 2022) that estimate the density or cumulative distribution function to construct prediction sets. As a reminder, in our setting, we only need access to the random samples generated from the fitted model instead of explicit density estimation. So our setting is more generic and applicable to real applications. Other than density estimation, Tumu et al. (2024) fits shape functions (convex hull, hyperrectangle, and ellipsoid) on the calibration data to provide a discontinuous prediction set as a union of fitted shapes. In Guha et al. (2024), the authors convert regression to a classification problem and can provide discontinuous prediction sets with interpolation of classification scores.

Efficiency, measured by the size of the prediction set, is a fundamental assessment of the quality of a CP method. Some works explore the effect of different non-conformity score measures such as

by scaling and reweighting the non-conformity score of each calibration data (Papadopoulos et al., 2002; 2011; Kivaranovic et al., 2020; Bellotti, 2020; Amoukou & Brunel, 2023). On the other hand, some works (Stutz et al., 2021; Einbinder et al., 2022) directly reduce inefficiency by incorporating a differentiable inefficiency measurement into the training loss. Bai et al. (2022) reformulate CP as a constrained optimization problem. By introducing an extra parameter and enriching the search space of the optimization problem, it demonstrates better performance in efficiency. Our main idea is partially motivated by this reformulation. We also gain insights from the recent development of CP for multi-target regression tasks and the definition of multivariate quantile function as discussed in (Messoudi et al., 2021; Sun & Yu, 2024). Recent advancements, such as probabilistic conformal prediction (PCP) using conditional random samples (Wang et al., 2023), have paved the way for CP in generative models. PCP leverages generated random samples to construct the uncertainty sets in the form of a union of balls around each generated sample, offering a discontinuous uncertainty set. This is a fundamental step towards better coverage efficiency, as ideally, we want the prediction set to avoid low-density regions. However, PCP faces two major limitations: ($i$) PCP does not distinguish different generated samples and assigns the same radius to each uncertainty ball, ($ii$) when the number of generated samples is relatively small, PCP can yield conservative prediction sets.

## 2 PRELIMINARIES

Conformal Prediction (CP) is a general framework for uncertainty quantification that provides statistically valid prediction regions for any predictive model under the mild assumption of data exchangeability. For any test input $X_{\text{test}}$, CP aims to construct a prediction set $\hat{\mathcal{C}}(X_{\text{test}})$ that satisfies the coverage guarantee

$$\Pr\big(Y_{\text{test}} \in \hat{\mathcal{C}}(X_{\text{test}})\big) \geq 1 - \alpha, \tag{1}$$

for a user-specified miscoverage rate $\alpha \in (0, 1)$.

Given calibration dataset $\mathcal{D} = \{(X_i, Y_i)\}_{i=1}^n$ with $X_i \in \mathcal{X}$ are features and $Y_i \in \mathcal{Y}$ are responses, CP defines a *non-conformity score* $E_i$ that measures how each sample conforms to fitted model $\hat{f}$. In regression, a common choice is the prediction error $E_i = \|Y_i - \hat{f}(X_i)\|$. The $\lceil (n+1)(1-\alpha) \rceil$-th order statistics of the calibration scores $\{E_i\}$ yields a threshold $\tau$, and the prediction set is

$$\hat{\mathcal{C}}(X_{\text{test}}) = \{y : \|y - \hat{f}(X_{\text{test}})\| \leq \tau\}.$$

The validity of (1) requires only that $(X_{\text{test}}, Y_{\text{test}})$ be exchangeable with $\mathcal{D}$.

*Prediction Set Efficiency.* Beyond validity, *efficiency*—the size of $\hat{\mathcal{C}}(X_{\text{test}})$—is crucial. Trivial sets covering the entire $\mathcal{Y}$ space ensure coverage but lack practical value. The aim is therefore to construct sets that are both valid and as small as possible.

## 3 PROPOSED METHOD

In this section, we introduce *generative conformal prediction with **O**ptimized **R**anking and **C**overage **A**llocation (ORCA)*, a three-stage framework that leverages the power of generative models to capture the full conditional distribution and employs an optimization-based, efficiency-oriented design of nonconformity scores to guarantee coverage validity while achieving superior efficiency.

Generative models provide richer information than classical predictors: instead of producing a point estimate or a few summary statistics (e.g., means or quantiles), they generate samples from the conditional distribution $\hat{p}(Y \mid X)$. Existing generative conformal methods typically collapse these samples into a single statistic (e.g., the minimum distance to the observation), discarding distributional structure and producing inefficient uncertainty sets. In contrast, our method vectorizes sample distances, capturing the distributional density information via ranking and enabling expressive exploration of the conditional distribution. At a high level, our uncertainty set takes the form of a *union of rank-dependent regions*, parameterized by a vector of radii $R(\alpha) \in \mathbb{R}^K$:

$$\hat{\mathcal{C}}_{\text{ORCA}}(X_{\text{test}}) = \bigcup_{r=1}^{K} \hat{\mathcal{C}}_r(R_r(\alpha)), \tag{2}$$

where $R_r(\alpha)$ is the $r$-th entry of $R(\alpha)$, and each $\hat{\mathcal{C}}_r(R_r(\alpha))$ is a region centered at the $r$-th ranked sample with radius $R_r(\alpha)$. When $Y \in R$, these regions reduce to intervals, and when $Y \in \mathbb{R}^d$, they correspond to Euclidean balls. Concretely, our method proceeds in three stages, with the complete

---

**Algorithm 1 ORCA**

---

1: **Input**: generative model $\hat{p}(Y|X)$; splits $\mathcal{D}_{\text{explore}}, \mathcal{D}_{\text{calib}}$; test feature $X_{\text{test}}$; miscoverage $\alpha$; sample size $K$
2: **Stage 1: Exploration with Sampling and Ranking**
3: **for** $(X_i, Y_i) \in \mathcal{D}_{\text{explore}}$ **do**
4:     Sample $\{\hat{Y}_{i,k}\}_{k=1}^{K} \sim \hat{p}(\cdot|X_i)$
5:     Rank samples by $m$-NN density (Eq. (3))
6:     Construct distance vector $E_i$ via Eq. (4)
7: **end for**
8: **Stage 2: Quantile Optimization**
9: Solve optimization problem (5) (via MILP (6)) to obtain threshold vector $T(\alpha)$
10: **Stage 3: Calibration**
11: **for** $(X_i, Y_i) \in \mathcal{D}_{\text{calib}}$ **do**
12:     Compute nonconformity scores $S_i$ (Eq. (7))
13: **end for**
14: Calibrate coverage threshold vector $R(\alpha) = t_\alpha T(\alpha)$ (Eq.(8))
15: **Prediction:** Draw $\{\hat{Y}_{\text{test},k}\}_{k=1}^{K} \sim \hat{p}(Y|X_{\text{test}})$ and rank, form $\hat{\mathcal{C}}_{\text{ORCA}}(X_{\text{test}})$ via Eq. (9).

---

procedure summarized in Algorithm 1. We set aside a dataset $\mathcal{D} = \{(X_i, Y_i)\}_{i=1}^{|\mathcal{D}|}$, distinct from the training data, and partition it into two disjoint parts: $\mathcal{D} = \mathcal{D}_{\text{explore}} \cup \mathcal{D}_{\text{calib}}$, with $|\mathcal{D}_{\text{explore}}| = n_1$ and $|\mathcal{D}_{\text{calib}}| = n_2$, referred to as the *exploration split* and the *calibration split*, respectively.

- **Stage 1 (Distribution Exploration with Sampling and Ranking).** For each observation in $\mathcal{D}_{\text{explore}}$, we draw multiple samples from the generative model and form a distance vector between the true label and the generated samples. The entries are then ranked by their average nearest-neighbor distances among generated samples, serving as a proxy for local density and prioritizing high-density regions. This exploration step preserves the full geometry of the predictive distribution.

- **Stage 2 (Optimal Coverage Allocation).** Given the ranked distance vectors, our goal is to identify a threshold vector that ensures valid coverage while maximizing efficiency. In the vector quantile setting, the coverage constraint admits multiple feasible threshold vectors, we cast the selection of an efficiency-oriented threshold vector as an optimization problem. We further reformulate this problem as a mixed-integer linear program (MILP), which offers both theoretical optimality guarantees and practical scalability. This optimization step is crucial, as it mirrors the oracle density level set that represents the ideal allocation of coverage.

- **Stage 3 (Conformal Calibration).** On $\mathcal{D}_{\text{calib}}$, we calibrate the optimized threshold vector obtained from Stage 2 using the split-conformal method. This ensures valid coverage at the desired level $1 - \alpha$ while preserving the efficiency gains during exploration.

## 3.1 STAGE-1: EFFICIENCY-ORIENTED DISTRIBUTIONAL QUANTILE EXPLORATION

The goal of Stage 1 is to explore the geometry of the predictive distribution. It produces rank-dependent distance vectors by sampling from the fitted model and ranking the generated samples by local density proxies. Intuitively, these vectors fully preserve distributional structure, including high- and low-density regions as well as multimodality.

**Sampling and Ranking.** For each data point $(X_i, Y_i) \in \mathcal{D}_{\text{explore}}$, we draw $K$ conditional samples $\{\hat{Y}_{i,k}\}_{k=1}^{K} \sim \hat{p}(\cdot \mid X_i)$. We then compute the average $m$-nearest-neighbor distance for each sample $\hat{Y}_{i,k}$, denoted $\bar{D}_{i,k}$, and rank the samples by distances:

$$\bar{D}_{i,j(1)} \leq \cdots \leq \bar{D}_{i,j(K)}, \tag{3}$$

where $j(r)$ indexes the sample with the $r$-th smallest $m$-nearest-neighbor distance. We then form a *ranked distance vector*

$$E_i = (E_{i,1}, \ldots, E_{i,K}), \quad E_{i,r} = \|Y_i - \hat{Y}_{i,j(r)}\|, \tag{4}$$

where each entry records the distance between the observed label $Y_i$ and the $r$-th ranked sample.

The rationale of ranking and constructing the ranked distance vector is two-fold: $(i)$ By retaining a full vector rather than a single scalar, we capture a richer structure of the model predictive distribution. $(ii)$ The ranking stratifies distributional structure: small $r$ emphasizes high-density regions where the model is confident, while large $r$ reflects sparser regions, highlighting uncertainty and tail behavior.

## 3.2 STAGE 2 OPTIMAL COVERAGE ALLOCATION

The goal of Stage 2 is to identify the optimal coverage structure and allocation across ranks by optimizing thresholds over the distance vectors from Stage 1. This efficiency-oriented optimization produces a rank-dependent threshold vector that will be calibrated in Stage 3 to ensure finite-sample validity. Intuitively, this optimization acts as a proxy for the oracle density level set, concentrating coverage in high-density regions while avoiding inefficient expansion into low-density areas.

**Quantile Optimization.** As described in (2), a key feature of our construction is that the ball radii are *rank-dependent*, meaning we allow each rank $r$ to adapt its radius to the local density structure. Concretely, for all samples at rank $r$, we seek a threshold $Q_r(\beta_r)$ defined as the $(1 - \beta_r)$ empirical quantile of $\{E_{i,r}\}_{i=1}^{n_1}$. Equivalently, learning the radii across ranks amounts to learning a quantile vector $\beta = (\beta_1, \ldots, \beta_K)^\top$.

Our objective is to minimize the overall size of the uncertainty set while satisfying the coverage constraint. Because the exact set size $\left| \cup_{r=1}^K \hat{\mathcal{C}}_r(Q_r(\beta_r)) \right|$ is analytically intractable due to overlaps across regions, we employ a tractable proxy: the sum of individual ball volumes,

$$\sum_{r=1}^K (Q_r(\beta_r))^d \propto \sum_{r=1}^K \left| \hat{\mathcal{C}}_r(Q_r(\beta_r)) \right| \geq \left| \cup_{r=1}^K \hat{\mathcal{C}}_r(Q_r(\beta_r)) \right|,$$

where $d = \dim(Y)$. Coverage is enforced by

$$\frac{1}{n_1} \sum_{i=1}^{n_1} \hat{S}_i(\beta) \geq 1 - \alpha, \qquad \hat{S}_i(\beta) = \max_{r \in [K]} \mathbb{1}\{E_{i,r} \leq Q_r(\beta_r)\}.$$

The resulting optimization problem is

$$\min_\beta \sum_{r=1}^K (Q_r(\beta_r))^d \quad \text{s.t.} \quad \frac{1}{n_1} \sum_{i=1}^{n_1} \hat{S}_i(\beta) \geq 1 - \alpha. \tag{5}$$

Intuitively, to minimize set size, the optimizer concentrates coverage in high-density regions, where small radii capture substantial probability mass. In contrast, achieving the same coverage in low-density regions requires much larger radii and is therefore inefficient. In doing so, it effectively mirrors the oracle level set, with $\beta$ acting as a rank-dependent allocation rule that shapes the uncertainty set toward the optimal density level set.

**Equivalent MILP Reformulation.** While the optimization problem (5) is conceptually simple, solving it directly by exhaustive search requires evaluating $n_1^K$ possible threshold configurations, which is computationally prohibitive. To address this, we reformulate the problem as a mixed integer linear program (MILP). This reformulation substantially enhances the scalability of our method. Theoretically, it is equivalent to the original problem and admits global optimality certificates from modern MILP solvers. Empirically, solving the MILP is often $10$–$100\times$ faster than heuristic approximation algorithms, while achieving the exact optimal solution.

For each rank $r \in \{1, \ldots, K\}$, let $E_{(\ell),r}$ denote the $\ell$-th order statistic of the set $\{E_{i,r}\}_{i=1}^{n_1}$, so that $E_{(1),r} \leq E_{(2),r} \leq \cdots \leq E_{(n_1),r}$. These serve as the candidate thresholds for $Q_r(\beta_r)$. We define incremental costs

$$c_{r,\ell} := E_{(\ell),r}^d - E_{(\ell-1),r}^d, \qquad \ell = 2, \ldots, n_1,$$

with $c_{r,1} := E_{(1),r}^d$, and coverage indicators

$$a_{i,r,\ell} := \mathbb{1}\{E_{i,r} \leq E_{(\ell),r}\}, \qquad i = 1, \ldots, n_1, \ \ell = 1, \ldots, n_1.$$

Let $\tau = \lceil n_1(1 - \alpha) \rceil$ denote the required number of points to cover. We then introduce two sets of binary variables: $(i)$ $z_{r,\ell} = 1$ if $E_{(\ell),r}$ is covered by the chose threshold and 0 otherwise; by construction, if $z_{r,\ell} = 1$ then all lower levels $\{1, \ldots, \ell - 1\}$ must also be 1; $(ii)$ $y_i = 1$ if sample $i$ is

covered and 0 otherwise. The MILP problem can then be rewritten as:

$$\min_{z,y} \quad \sum_{r=1}^{K} \sum_{\ell=1}^{n_1} c_{r,\ell}\, z_{r,\ell} \tag{6a}$$

$$\text{s.t.} \quad y_i \leq \sum_{r=1}^{K} \sum_{\ell=1}^{n_1} a_{i,r,\ell}\, z_{r,\ell} \ (\forall i \in [n_1]), \quad \sum_{i=1}^{n_1} y_i \geq \tau \tag{6b}$$

$$z_{r,\ell} \geq z_{r,\ell+1} \quad (\forall r \in [K],\ \ell \in [n_1-1]), \quad z_{r,\ell},\, y_i \in \{0,1\} \tag{6c}$$

*Remark* 3.1 (Exact MILP Reformulation of (5)). Let $\ell_r^*$ denote the choosen threshold at rank $r$ with order $l^*$. Then by summing increments,

$$\sum_{r=1}^{K} \sum_{\ell=1}^{n_1} c_{r,\ell}\, z_{r,\ell} = \sum_{r=1}^{K} \sum_{\ell \leq \ell_r^*} \left( E_{(\ell),r}^d - E_{(\ell-1),r}^d \right) = \sum_{r=1}^{K} E_{(\ell_r^*),r}^d.$$

We can observe the MILP objective (6a) coincides with $\sum_{r=1}^{K} (Q_r(\beta_r))^d$ in (5). Constraint (6b) encodes the coverage guarantee that at least $\tau$ points must be covered. Constraint (6c) enforces monotonicity across thresholds and integrality of the decision variables. Hence the MILP (6) is an *exact reformulation* of (5) with identical optimal value. This directly yields Theorem 3.2, with proof deferred to Appendix D.

**Theorem 3.2** (Exact MILP Reformulation and Optimality). *The MILP formulation (6) is equivalent to optimization (5). In particular, there exists a one-to-one correspondence between feasible quantile vectors $\beta = (\beta_1, \ldots, \beta_K)$ in (5) and feasible binary assignments $(z,y)$ in (6). Consequently, the optimal solution of the MILP yields a globally optimal solution to the original problem (5).*

### 3.3 Stage-3: Conformal Calibration with Coverage Guarantees

In Stage 3, we use $\mathcal{D}_{\text{calib}}$ to calibrate the resulting threshold vector from Stage 2, denoted as $T(\alpha) \in \mathbb{R}^K$, whose $r$-th component is denoted $T_r(\alpha)$. Define the nonconformity scores

$$S_i := \min_{r \in [K]} \frac{E_{i,r}}{T_r(\alpha)}, \qquad i \in \mathcal{D}_2. \tag{7}$$

The score quantify how tightly each calibration point is covered relative to the Stage 1 threshold: $S_i < 1$ indicates over-coverage and $S_i > 1$ under-coverage. We then obtain the final coverage radii $R(\alpha) \in \mathbb{R}^K$, with $R_r(\alpha)$ denoting its $r$-th component, by scaling the Stage 1 thresholds $T(\alpha)$ using a conformal factor $t_\alpha$:

$$R(\alpha) := t_\alpha\, T(\alpha), \quad t_\alpha := \min \left\{ \tau \in \mathbb{R} : \frac{1}{n_2+1} \sum_{i=1}^{n_2} \mathbb{1}\{S_i \leq \tau\} \geq 1-\alpha \right\}. \tag{8}$$

The resulting prediction set takes the form

$$\hat{\mathcal{C}}_{\text{ORCA}}(X_{\text{test}}) = \bigcup_{r=1}^{K} \hat{\mathcal{C}}_r(R_r(\alpha)) = \bigcup_{r=1}^{K} \left\{ y : \|y - \hat{Y}_{\text{test},j(r)}\| \leq R_r(\alpha) \right\}. \tag{9}$$

**Theorem 3.3** (Validity of ORCA). *Suppose the test point $(X_{test}, Y_{test})$ is exchangeable with the calibration data $\mathcal{D}_{calib}$. Then the prediction set $\hat{\mathcal{C}}_{ORCA}(X_{test})$ satisfies*

$$\Pr\left( Y_{test} \in \hat{\mathcal{C}}_{ORCA}(X_{test}) \right) \geq 1-\alpha.$$

The proof follows from the standard conformal prediction argument with the exchangeability assumption. The full proof is deferred to Appendix D.

### 3.4 Asymptotics and Level-Set Limit

We now formalize the asymptotic properties of the proposed method, and all the proofs and supporting lemmas are deferred to Appendix D.

**Proposition 3.4** (Density–Order Consistency with Averaged $m$-NN Distances). *For any given $x$, let $Z_1, \ldots, Z_K \overset{i.i.d.}{\sim} p(\cdot \mid x)$ for a density $p(\cdot \mid x)$ on a compact domain $\mathcal{Y} \subset \mathbb{R}^d$. Assume $p(\cdot \mid x)$ is continuous and bounded away from 0 and $\infty$. For $z \in \mathcal{Y}$, let $\bar{R}_m(z) = \frac{1}{m} \sum_{j=1}^{m} R_j(z)$, where $R_j(z)$ is the Euclidean distance from $z$ to its $j$-th nearest neighbor among $\{Z_1, \ldots, Z_K\}$. Suppose $m \to \infty$ and $m/K \to 0$ as $K \to \infty$. Then ordering $\{Z_j\}$ by increasing $\bar{R}_m(\cdot)$ coincides with ordering by decreasing $p(\cdot \mid x)$, with probability $1 - o(1)$ :*

$$p\big(Z_{(1)} \mid x\big) \geq p\big(Z_{(2)} \mid x\big) \geq \cdots \geq p\big(Z_{(K)} \mid x\big),$$

*where $Z_{(r)}$ is the point with the $r$-th smallest $\bar{R}_m(\cdot)$.*

**Theorem 3.5** (Convergence to the Optimal HDR). *Under the assumptions of Proposition 3.4, let $\mathcal{C}_{K,n_1,n_2}(x)$ denote the two-stage predictor constructed from $K$ generated samples, with $|\mathcal{D}_{explore}| = n_1$, and $|\mathcal{D}_{calib}| = n_2$. If $K, n_1, n_2 \to \infty$, then with probability $1 - o(1)$,*

$$\mu(\mathcal{C}_{K,n_1,n_2}(x) \triangle L_{\tau^\star}(x)) \to 0,$$

*where $L_{\tau^\star}(x) := \{y : p(y \mid x) \geq \tau^\star(x)\}$ is the $(1 - \alpha)$ highest-density region, and $\tau^\star(x)$ is the largest cutoff satisfying $\int_{L_{\tau^\star}(x)} p(y \mid x) \, dy = 1 - \alpha$. Hence, the Lebesgue measure of the symmetric difference between $\mathcal{C}_{K,n_1,n_2}(x)$ and the oracle HDR $L_{\tau^\star}(x)$ converges to zero in probability.*

Proposition 3.4 shows that the ranking is consistent with the underlying density ordering: points in higher-density regions admit systematically smaller averaged $m$-NN radii, so sorting by $\bar{R}_m$ recovers the true density order. Theorem 3.5 then establishes the asymptotic consequence: as $K, n_1, n_2 \to \infty$, the optimization-based coverage allocation concentrates precisely on these high-density regions. Consequently, the resulting uncertainty set converges to the oracle highest-density region.

## 4 EXPERIMENTS

In this section, we evaluate the empirical performance of ORCA compared with several state-of-the-art baselines on 6 synthetic data, a semi-synthetic data, and 9 real-world datasets. We demonstrate the superior performance of ORCA, especially in handling complex distributions.

**Baselines:** We compare against state-of-the-art conformal prediction approaches: Conformalized Quantile Regression (CQR) (Romano et al., 2019), Conformal Histogram Regression (CHR) (Sesia & Romano, 2021), Distributional Conformal Prediction (DCP) (Chernozhukov et al., 2021), DistSplit and CDSplit (Izbicki et al., 2020), Probabilistic Conformal Prediction (PCP) (Wang et al., 2023), Conformal Region Designer (CRD) (Tumu et al., 2024), and Regression-to-Classification Conformal Prediction (R2CCP) (Guha et al., 2024). CQR and CHR produce continuous prediction sets by estimating conditional quantiles and conditional histograms, respectively. DCP, DistSplit, and CDSplit integrate estimated distribution/density functions within the conformal framework. The remaining methods generate discontinuous prediction sets: R2CCP converts the regression problem to classification and interpolates classification scores, while CRD optimizes geometric shapes (hyperrectangles, ellipsoids, and convex hulls) to minimize volume. The implementations of CQR, CHR, R2CCP, CRD, and PCP are based on the authors' public repositories, while DCP, DistSplit, and CDSplit implementations are based on a public repository Wang et al. (2023).

**Implementation Configuration:** Through experiments, we set $\alpha = 0.1$ and evaluate methods using marginal coverage (Cover), worst-slab conditional coverage (Cond_Cover), and prediction set size (Eff). We set $K = 20$ for experiments (except for $K = 50$ for the bike dataset). We set $m = \lceil \sqrt{K} \rceil$ for $m$-nearest neighbor distance calculations, a practical choice commonly adopted in nearest neighbor methods. For baseline model and conditional sample generation, we adopt the quantile regression forests (Meinshausen, 2006) and mixture density networks (Bishop, 1994; Rothfuss et al., 2019). Further details of the implementation of each method, synthetic data generation, and modeling procedure can be found in Appendix E.

### 4.1 SYNTHETIC DATA

We benchmark on six synthetic datasets (Circles, Moon, S-shape, Spiral, Roll, Mixture Gaussian) with multimodal distributions. Models are trained on 3,000 points; we use 500 for exploration, 500 for calibration, and 1,000 for testing. Table 1 reports averages over 30 runs. All methods achieve nominal coverage, but ORCA consistently attains the smallest sets. Visualizations (Appendix F.1) show

Table 1: Summary of synthetic data results.

| Dataset | Metric | CQR | CHR | DCP | DistSplit | CDSplit | CRD | R2CCP | PCP | ORCA |
|---|---|---|---|---|---|---|---|---|---|---|
| Circles | Cover | 0.90 ± 0.01 | 0.90 ± 0.01 | 0.90 ± 0.01 | 0.90 ± 0.01 | 0.91 ± 0.01 | 0.90 ± 0.01 | 0.90 ± 0.01 | 0.90 ± 0.01 | 0.90 ± 0.01 |
| | Cond_Cover | 0.85 ± 0.04 | 0.84 ± 0.05 | 0.86 ± 0.03 | 0.86 ± 0.03 | 0.87 ± 0.03 | 0.80 ± 0.03 | 0.86 ± 0.01 | 0.85 ± 0.03 | 0.86 ± 0.03 |
| | Eff | 1.70 ± 0.01 | 1.49 ± 0.06 | 1.64 ± 0.05 | 1.64 ± 0.05 | 0.72 ± 0.04 | 1.61 ± 0.01 | 2.77 ± 0.04 | 0.68 ± 0.31 | **0.63 ± 0.24** |
| Moon | Cover | 0.90 ± 0.01 | 0.90 ± 0.01 | 0.90 ± 0.01 | 0.90 ± 0.01 | 0.92 ± 0.01 | 0.90 ± 0.01 | 0.90 ± 0.01 | 0.90 ± 0.01 | 0.90 ± 0.02 |
| | Cond_Cover | 0.69 ± 0.05 | 0.69 ± 0.08 | 0.77 ± 0.06 | 0.75 ± 0.06 | 0.87 ± 0.05 | 0.78 ± 0.03 | 0.86 ± 0.02 | 0.85 ± 0.03 | 0.85 ± 0.04 |
| | Eff | 0.69 ± 0.05 | 0.69 ± 0.08 | 0.77 ± 0.06 | 0.75 ± 0.06 | 0.42 ± 0.03 | 0.33 ± 0.03 | 1.84 ± 0.11 | 0.35 ± 0.04 | **0.29 ± 0.04** |
| S-shape | Cover | 0.90 ± 0.01 | 0.90 ± 0.01 | 0.90 ± 0.01 | 0.90 ± 0.01 | 0.91 ± 0.02 | 0.90 ± 0.01 | 0.90 ± 0.01 | 0.90 ± 0.02 | 0.90 ± 0.01 |
| | Cond_Cover | 0.53 ± 0.04 | 0.80 ± 0.07 | 0.80 ± 0.06 | 0.80 ± 0.07 | 0.88 ± 0.02 | 0.81 ± 0.04 | 0.85 ± 0.02 | 0.84 ± 0.03 | 0.83 ± 0.04 |
| | Eff | 3.36 ± 0.02 | 3.34 ± 0.02 | 3.34 ± 0.02 | 3.34 ± 0.02 | 1.16 ± 0.11 | 2.63 ± 0.25 | 6.94 ± 0.47 | 0.58 ± 0.01 | **0.51 ± 0.05** |
| Spiral | Cover | 0.90 ± 0.01 | 0.90 ± 0.01 | 0.90 ± 0.01 | 0.90 ± 0.01 | 0.91 ± 0.01 | 0.90 ± 0.01 | 0.90 ± 0.01 | 0.90 ± 0.01 | 0.90 ± 0.01 |
| | Cond_Cover | 0.75 ± 0.06 | 0.81 ± 0.06 | 0.82 ± 0.07 | 0.80 ± 0.06 | 0.87 ± 0.04 | 0.81 ± 0.05 | 0.87 ± 0.02 | 0.85 ± 0.04 | 0.85 ± 0.05 |
| | Eff | 17.59 ± 0.40 | 17.02 ± 0.58 | 16.94 ± 0.55 | 16.92 ± 0.53 | 10.67 ± 0.98 | 12.62 ± 1.27 | 45.26 ± 2.83 | 6.27 ± 0.68 | **5.96 ± 0.57** |
| Roll | Cover | 0.90 ± 0.01 | 0.90 ± 0.01 | 0.90 ± 0.01 | 0.90 ± 0.01 | 0.91 ± 0.01 | 0.90 ± 0.01 | 0.90 ± 0.01 | 0.90 ± 0.01 | 0.90 ± 0.02 |
| | Cond_Cover | 0.65 ± 0.07 | 0.81 ± 0.05 | 0.77 ± 0.05 | 0.76 ± 0.06 | 0.84 ± 0.04 | 0.71 ± 0.05 | 0.86 ± 0.03 | 0.84 ± 0.04 | 0.83 ± 0.04 |
| | Eff | 18.59 ± 0.17 | 15.33 ± 0.42 | 17.39 ± 0.49 | 17.33 ± 0.47 | 8.50 ± 0.48 | 14.64 ± 0.92 | 36.51 ± 2.56 | 4.67 ± 0.57 | **4.46 ± 0.58** |
| Mixture Gaussian | Cover | 0.90 ± 0.01 | 0.90 ± 0.01 | 0.90 ± 0.01 | 0.90 ± 0.01 | 0.91 ± 0.01 | 0.90 ± 0.01 | 0.90 ± 0.02 | 0.90 ± 0.01 | 0.90 ± 0.02 |
| | Cond_Cover | 0.89 ± 0.01 | 0.89 ± 0.04 | 0.89 ± 0.04 | 0.89 ± 0.05 | 0.87 ± 0.03 | 0.89 ± 0.03 | 0.82 ± 0.03 | 0.85 ± 0.03 | 0.86 ± 0.03 |
| | Eff | 12.61 ± 0.11 | 12.66 ± 0.10 | 12.62 ± 0.11 | 12.61 ± 0.10 | **6.89 ± 0.19** | 13.64 ± 0.43 | 14.49 ± 0.80 | 7.84 ± 0.53 | 7.66 ± 0.58 |

[1] Metrics include marginal coverage (Cover), worst-slab conditional coverage (Cond_Cover), and prediction set efficiency (Eff). Values report mean ± standard error over 30 runs. The lowest efficiency per dataset is bolded and highlighted along with corresponding coverage metrics.

that PCP and ORCA produce discontinuous sets suited to multimodality; ORCA further improves efficiency via locally adaptive radii—expanding in high-density regions and contracting in sparse areas—enabled by its vectorized scores and optimized coverage allocation.

## 4.2 MNIST: Digit Image Generation

We design a semi-synthetic MNIST experiment (Deng, 2012) to assess ORCA on high-dimensional data, where interpreting uncertainty sets is challenging. Given a label $X \in \{0, \ldots, 9\}$, the target $Y$ is an image of digit $X$. We train a VAE encoder and construct uncertainty sets in its latent space. To ensure independence, we hold out a disjoint subset to serve as an *external sampler*: for any $X$, it returns images of digit $X$ unseen during training/exploration/calibration. These samples are embedded by the trained encoder and used by PCP and ORCA.

Figure 2 shows samples from PCP and ORCA prediction sets with $K = 40$ for different $X$ (rows). Both methods achieve nominal coverage, but ORCA samples lie closer to the target image, indicating tighter sets. Since raw set size in image space is not directly meaningful, we thus report the *off-class rate (OCR)*—the probability a non-$X$ digit appears in the set. ORCA attains $0.16 \pm 0.02$ OCR (mean±SE over 30 runs), versus $0.22 \pm 0.03$ for PCP.

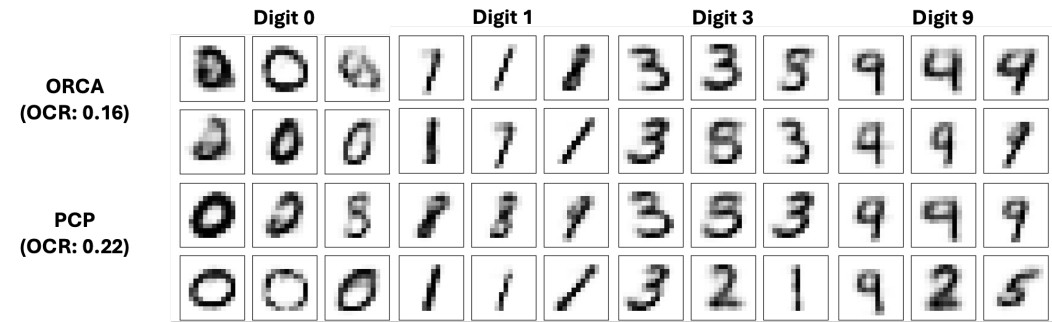

Figure 2: Comparison images sampled from uncertainty sets generated by ORCA (top rows) and PCP (bottom rows) for different target digits. ORCA produces more semantically consistent samples with a lower off-class rate (OCR), achieving 0.16 versus 0.22 for PCP. This demonstrates that ORCA yields more efficient prediction sets that more accurately reflect the target digits.

## 4.3 Real Data

We evaluate ORCA on popular benchmarks—Blog (Buza, 2014b), Facebook Comment Volume (Fb1, Fb2) (Singh, 2015b), MEPS (years 19–21), Protein Tertiary Structure (Bio), and Temperature Forecasts (Cho et al., 2020)—and on a NYC bike-sharing dataset with a 2D destination prediction task. For each dataset, we use 500 samples for exploration, 500 for calibration, 1,000 for testing, and the remainder for training. Here, we observe that DCP is unstable on real datasets (as also noted by Sesia & Romano (2021)). We therefore replace it with a hybrid variant, DCP-CQR, which stabilizes DCP using CQR.

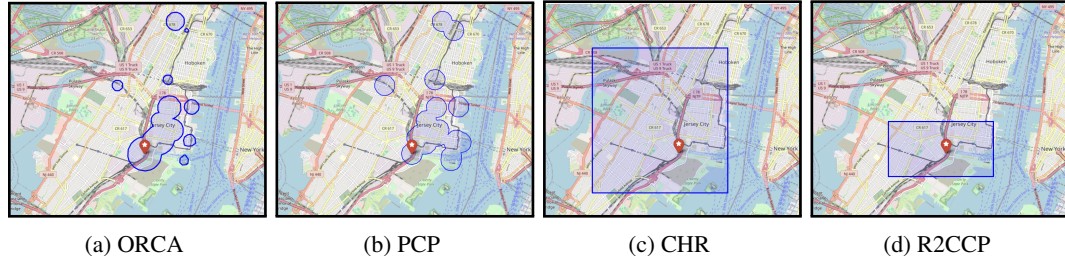

| (a) ORCA | (b) PCP | (c) CHR | (d) R2CCP |

Figure 3: Comparison of prediction sets across four methods that have the highest efficiency on the Bike dataset. The colored area indicates the prediction set, and the red marker is the target location.

Table 2 shows that all methods meet nominal coverage, while ORCA consistently yields smaller prediction sets. The gains are especially pronounced on complex data (e.g., Blog with 280 features), where the conditional distribution $P(Y \mid X)$ is highly structured and benefits more from ORCA's rank-dependent coverage allocation: high-density regions receive larger local radii, whereas low-density regions are down-weighted. In contrast, methods that impose a single global scale or simple geometry tend to over-cover sparse areas to maintain coverage, increasing set size. Figure 3 illustrates NYC bike destination prediction. PCP and ORCA produce discontinuous regions (unions of local balls), while CHR and R2CCP return rectangle-shaped continuous sets. The discontinuity is desirable here: destinations cluster around hubs, with pockets of low density elsewhere. ORCA adapts to this heterogeneity—expanding near stations and shrinking to near-zero in sparse zones—thereby improving efficiency without sacrificing coverage. Compared to PCP, ORCA's optimized coverage allocation across ranks systematically reduces unnecessary expansion in low-density areas, leading to tighter and more targeted sets.

Table 2: Summary of real data results.

| Dataset | Metric | CQR | CHR | DCP-CQR | DistSplit | CDSplit | CRD | R2CCP | PCP | **ORCA** |
|---|---|---|---|---|---|---|---|---|---|---|
| Fb1 | Cover | $0.90 \pm 0.00$ | $0.90 \pm 0.01$ | $0.90 \pm 0.01$ | $0.90 \pm 0.00$ | $0.92 \pm 0.01$ | $0.90 \pm 0.01$ | $0.90 \pm 0.01$ | $0.90 \pm 0.01$ | $0.90 \pm 0.02$ |
| | Cond_Cover | $0.87 \pm 0.02$ | $0.86 \pm 0.05$ | $0.88 \pm 0.00$ | $0.89 \pm 0.03$ | $0.87 \pm 0.07$ | $0.83 \pm 0.03$ | $0.76 \pm 0.04$ | $0.82 \pm 0.03$ | $0.80 \pm 0.05$ |
| | Eff | $14.78 \pm 2.31$ | $11.23 \pm 1.34$ | $13.73 \pm 1.55$ | $15.88 \pm 2.07$ | $35.09 \pm 2.69$ | $54.07 \pm 40.50$ | $12.97 \pm 0.82$ | $7.79 \pm 0.28$ | $\mathbf{3.70 \pm 1.75}$ |
| Fb2 | Cover | $0.90 \pm 0.01$ | $0.89 \pm 0.01$ | $0.90 \pm 0.02$ | $0.90 \pm 0.01$ | $0.93 \pm 0.01$ | $0.90 \pm 0.02$ | $0.90 \pm 0.02$ | $0.90 \pm 0.01$ | $0.90 \pm 0.01$ |
| | Cond_Cover | $0.88 \pm 0.02$ | $0.84 \pm 0.04$ | $0.89 \pm 0.05$ | $0.89 \pm 0.01$ | $0.87 \pm 0.08$ | $0.84 \pm 0.03$ | $0.84 \pm 0.03$ | $0.82 \pm 0.03$ | $0.80 \pm 0.04$ |
| | Eff | $18.31 \pm 1.57$ | $10.12 \pm 1.04$ | $10.54 \pm 1.12$ | $14.48 \pm 1.43$ | $44.41 \pm 1.61$ | $56.26 \pm 2.64$ | $13.39 \pm 1.65$ | $9.35 \pm 0.21$ | $\mathbf{5.68 \pm 1.62}$ |
| Blog | Cover | $0.90 \pm 0.02$ | $0.90 \pm 0.02$ | $0.90 \pm 0.01$ | $0.90 \pm 0.01$ | $0.96 \pm 0.01$ | $0.90 \pm 0.01$ | $0.89 \pm 0.02$ | $0.90 \pm 0.01$ | $0.90 \pm 0.01$ |
| | Cond_Cover | $0.87 \pm 0.02$ | $0.89 \pm 0.03$ | $0.90 \pm 0.04$ | $0.88 \pm 0.03$ | $0.91 \pm 0.02$ | $0.81 \pm 0.03$ | $0.74 \pm 0.04$ | $0.81 \pm 0.04$ | $0.82 \pm 0.05$ |
| | Eff | $16.18 \pm 2.61$ | $10.26 \pm 1.77$ | $11.37 \pm 1.76$ | $16.93 \pm 2.13$ | $39.95 \pm 3.30$ | $63.07 \pm 37.05$ | $10.95 \pm 1.82$ | $8.27 \pm 0.21$ | $\mathbf{3.61 \pm 1.27}$ |
| Bio | Cover | $0.90 \pm 0.01$ | $0.90 \pm 0.01$ | $0.90 \pm 0.01$ | $0.90 \pm 0.01$ | $0.90 \pm 0.01$ | $0.90 \pm 0.01$ | $0.89 \pm 0.01$ | $0.90 \pm 0.01$ | $0.90 \pm 0.01$ |
| | Cond_Cover | $0.90 \pm 0.03$ | $0.90 \pm 0.04$ | $0.89 \pm 0.03$ | $0.90 \pm 0.04$ | $0.86 \pm 0.03$ | $0.84 \pm 0.04$ | $0.83 \pm 0.03$ | $0.85 \pm 0.02$ | $0.82 \pm 0.03$ |
| | Eff | $13.09 \pm 0.18$ | $\mathbf{9.44 \pm 0.33}$ | $11.58 \pm 0.29$ | $11.73 \pm 0.30$ | $10.89 \pm 0.39$ | $22.10 \pm 2.79$ | $9.47 \pm 0.48$ | $11.54 \pm 0.54$ | $10.88 \pm 0.71$ |
| Meps 19 | Cover | $0.90 \pm 0.01$ | $0.90 \pm 0.01$ | $0.91 \pm 0.00$ | $0.90 \pm 0.01$ | $0.92 \pm 0.01$ | $0.90 \pm 0.01$ | $0.89 \pm 0.12$ | $0.90 \pm 0.01$ | $0.90 \pm 0.01$ |
| | Cond_Cover | $0.89 \pm 0.02$ | $0.90 \pm 0.03$ | $0.90 \pm 0.04$ | $0.86 \pm 0.03$ | $0.88 \pm 0.02$ | $0.83 \pm 0.03$ | $0.81 \pm 0.15$ | $0.85 \pm 0.03$ | $0.87 \pm 0.04$ |
| | Eff | $30.75 \pm 1.42$ | $17.41 \pm 1.23$ | $19.35 \pm 1.47$ | $29.91 \pm 2.11$ | $23.73 \pm 1.03$ | $65.66 \pm 23.88$ | $22.74 \pm 1.24$ | $16.95 \pm 0.66$ | $\mathbf{14.94 \pm 2.96}$ |
| Meps 20 | Cover | $0.90 \pm 0.01$ | $0.90 \pm 0.01$ | $0.91 \pm 0.01$ | $0.90 \pm 0.01$ | $0.92 \pm 0.01$ | $0.91 \pm 0.02$ | $0.88 \pm 0.02$ | $0.90 \pm 0.02$ | $0.90 \pm 0.01$ |
| | Cond_Cover | $0.90 \pm 0.02$ | $0.90 \pm 0.04$ | $0.89 \pm 0.00$ | $0.90 \pm 0.03$ | $0.85 \pm 0.03$ | $0.84 \pm 0.04$ | $0.80 \pm 0.04$ | $0.85 \pm 0.03$ | $0.85 \pm 0.03$ |
| | Eff | $28.27 \pm 1.76$ | $17.84 \pm 1.41$ | $18.73 \pm 1.72$ | $28.71 \pm 3.03$ | $23.81 \pm 1.22$ | $71.10 \pm 26.01$ | $19.46 \pm 1.59$ | $16.06 \pm 1.87$ | $\mathbf{14.02 \pm 2.86}$ |
| Meps 21 | Cover | $0.90 \pm 0.01$ | $0.90 \pm 0.01$ | $0.90 \pm 0.02$ | $0.90 \pm 0.01$ | $0.91 \pm 0.01$ | $0.90 \pm 0.01$ | $0.88 \pm 0.06$ | $0.90 \pm 0.01$ | $0.90 \pm 0.02$ |
| | Cond_Cover | $0.90 \pm 0.02$ | $0.90 \pm 0.03$ | $0.89 \pm 0.05$ | $0.89 \pm 0.02$ | $0.84 \pm 0.02$ | $0.83 \pm 0.03$ | $0.80 \pm 0.08$ | $0.86 \pm 0.03$ | $0.82 \pm 0.04$ |
| | Eff | $30.42 \pm 1.33$ | $17.89 \pm 1.35$ | $20.23 \pm 2.26$ | $29.42 \pm 2.13$ | $24.39 \pm 3.17$ | $53.65 \pm 18.49$ | $19.22 \pm 1.60$ | $16.93 \pm 1.32$ | $\mathbf{14.16 \pm 1.97}$ |
| Temperature | Cover | $0.90 \pm 0.02$ | $0.90 \pm 0.01$ | $0.90 \pm 0.01$ | $0.90 \pm 0.02$ | $0.90 \pm 0.01$ | $0.90 \pm 0.02$ | $0.90 \pm 0.01$ | $0.90 \pm 0.01$ | $0.90 \pm 0.01$ |
| | Cond_Cover | $0.89 \pm 0.04$ | $0.89 \pm 0.05$ | $0.88 \pm 0.03$ | $0.88 \pm 0.05$ | $0.85 \pm 0.03$ | $0.80 \pm 0.04$ | $0.79 \pm 0.04$ | $0.85 \pm 0.03$ | $0.82 \pm 0.03$ |
| | Eff | $3.45 \pm 0.10$ | $3.42 \pm 0.12$ | $3.43 \pm 0.10$ | $3.46 \pm 0.11$ | $2.59 \pm 0.07$ | $3.78 \pm 0.11$ | $2.55 \pm 0.09$ | $3.80 \pm 0.01$ | $\mathbf{2.43 \pm 0.13}$ |

[1] Metrics include marginal coverage (Cover), worst-slab conditional coverage (Cond_Cover), and prediction set efficiency (Eff). Values report mean $\pm$ standard error over 30 runs. The lowest efficiency per dataset is bolded and highlighted along with corresponding coverage metrics.

## 5 DISCUSSION AND FUTURE WORK

We propose ORCA, a conformal framework for generative models that leverages ranked generative samples and optimization-based coverage allocation to improve efficiency, yielding flexible (often discontinuous) uncertainty regions that concentrate in high-density areas while preserving marginal coverage. Our exact MILP reformulation and theory establish validity and asymptotic optimality, and experiments on synthetic and real data show consistent efficiency gains over strong baselines.

Several limitations point to future work. First, scalability: although the MILP is exact, solving it can be costly as $K$ and $n_1$ grow; lightweight relaxations, decomposition, or streaming updates are natural extensions. Second, high-dimensional geometry: Euclidean distances suffer from concentration and is less meaningful, so performance depends on operating in informative latent spaces; this motivates integrating learned metrics or pretrained embeddings for images and text. Finally, while calibration ensures coverage even under model misspecification, efficiency hinges on sample quality from the generative backend; coupling ORCA with diagnostics and adaptive allocation in rich modalities (e.g., diffusion or language models) is a promising direction.

**Ethics statement** This work does not involve human subjects, sensitive personal data, or applications that pose direct risks of harm. All datasets used are publicly available and widely adopted in the research community. We ensured that our methods were applied in a way that does not introduce or amplify unfair bias, discrimination, or privacy risks. The research was conducted in accordance with the ICLR Code of Ethics.

**Reproducibility statement** We have taken steps to ensure the reproducibility of our results. Experimental details, including dataset descriptions, preprocessing, model architectures, hyperparameters, and evaluation metrics, are documented in the main text and supplementary materials. An anonymized codebase with implementation and notebooks for reproducing our experiments will be made available to reviewers upon request.

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

## A  THE USE OF LARGE LANGUAGE MODELS (LLMS)

LLMs were used solely as a writing assistant for polishing and improving the clarity of our manuscript (e.g., refining grammar, rephrasing sentences for conciseness, and improving flow). LLMs were not used for generating research ideas, conducting analysis, deriving results, or writing original technical content. All conceptual contributions, methodological developments, proofs, and experiments were designed, implemented, and validated by the authors.

## B  AVERAGE $m$-NEAREST NEIGHBOR DISTANCE

The average $m$-nearest neighbor distance for a generated sample $\hat{Y}_{i,k}$ can be computed as follows:

1. (Optional) Sample a sequence of reference samples $\{\tilde{Y}_{i,j}\}_{l=1}^{M} \sim \hat{p}(\cdot|X_i)$ for ranking estimation. Otherwise, set $\tilde{Y}_{i,l} = \hat{Y}_{i,l}$ for $l = 1, \ldots, K$ and $M = K$.

2. Compute the pairwise distances of generated samples among reference samples:
$$D_i(k,l) := \|\hat{Y}_{i,k} - \tilde{Y}_{i,l}\|, \text{ for } k \in \{1, \ldots, K\}, \ l \in \{1, \ldots, M\}.$$

3. Sort the distances to $\hat{Y}_{i,k}$ such that:
$$D_i(k,(1)) \leq \cdots \leq D_i(k,(M)),$$
where $D_i(k,(l))$ denotes the $l$-th smallest distance to $\hat{Y}_{i,k}$.

4. Calculate average $m$-nearest neighbor distance:
$$\bar{D}_{i,k} = \frac{1}{m} \sum_{l=1}^{m} D_i(k,(l)).$$

## C  COMPARISON WITH PCP

Probabilistic Conformal Prediction (PCP) (Wang et al., 2023) first studied CP for generative models and utilized the information of random samples. In contrast to our method, which considers all the distances between each random sample and the true response, PCP defines its non-conformity score as the minimum distance between random samples and the true response:
$$E_i = \min_{k=1,\ldots,K} \|Y_i - \hat{Y}_{i,k}\|. \tag{10}$$
for each calibration data $(X_i, Y_i)$. For a new test data $(X_{n+1}, Y_{n+1})$, the prediction set of PCP is constructed as the union of $K$ regions:
$$\hat{\mathcal{C}}(X_{n+1}) = \cup_{k=1}^{K}\hat{\mathcal{C}}_k, \tag{11}$$
where each region $\hat{\mathcal{C}}_k$ is defined by
$$\hat{\mathcal{C}}_k = \left\{ y : \|y - \hat{Y}_{n+1,k}\| \leq Q(E_{1:n} \cup \{\infty\}; \alpha) \right\}.$$
Here $Q(\cdot; \alpha)$ is the $1 - \alpha$ quantile of the empirical distribution of the non-conformity scores $E_1, \ldots, E_n$.

As noted, PCP adopts a scalar non-conformity score, which assigns a uniform radius to all prediction regions $\hat{\mathcal{C}}_k$. This uniform treatment of random samples limits adaptability, as ideally, high-density regions should have larger radius, while low-density regions require smaller ones. In contrast, ORCA introduces a vectorized non-conformity score that captures the full complexity of the prediction error distribution. By ranking samples based on their empirical density, our method enables individualized optimization of the ball radius for each sample. As demonstrated in Figure 3, the ball radius for low-density regions nearly diminishes, highlighting the flexibility of our approach. Another limitation of PCP arises when $K$ is small. In such cases, the empirical distribution of minimum distances may become heavy-tailed, requiring a large quantile (radius) to maintain coverage, leading to an overly conservative prediction set $\hat{\mathcal{C}}(X_{n+1})$. In contrast, the proposed ORCA overcomes this by introducing an individualized quantile selection, relaxing the constraint for attaining the $1 - \alpha$ quantile uniformly across all samples at each rank and extensively reducing the impact of the distribution's tail.

# D    PROOF

## D.1    PROOF OF THEOREM 3.2

*Proof.* For each rank $r$, let $\{E_{(1),r} \leq \cdots \leq E_{(n_1),r}\}$ denote the order statistics of $\{E_{i,r}\}_{i=1}^{n_1}$, and define incremental costs

$$c_{r,1} = E_{(1),r}^d, \qquad c_{r,\ell} = E_{(\ell),r}^d - E_{(\ell-1),r}^d \ \ (\ell \geq 2).$$

**(i) From quantile problem to MILP.** Let $\beta$ be feasible for (5). For each $r$, let $\ell_r$ satisfy $Q_r(\beta_r) = E_{(\ell_r),r}$ and set $z_{r,\ell} := \mathbb{1}\{\ell \leq \ell_r\}$. By the definition of $y_i$, we have

$$y_i := \mathbb{1}\left\{\sum_{r=1}^{K}\sum_{\ell=1}^{n_1} a_{i,r,\ell}\, z_{r,\ell} \geq 1\right\} = \max_{r\in[K]} \mathbb{1}\{E_{i,r} \leq Q_r(\beta_r)\}.$$

Then:

*(a) Monotonicity (6c).* By choice of $\ell_r$ and construction $z_{r,l}$, $z_{r,\ell} \geq z_{r,\ell+1}$ for all $\ell$, and $z_{r,\ell} \in \{0,1\}$. Also $y_i \in \{0,1\}$ by definition.

*(b) Coverage quota (6b).* We have $y_i \leq \sum_{r,\ell} a_{i,r,\ell} z_{r,\ell}$ and

$$\sum_{i=1}^{n_1} y_i = \sum_{i=1}^{n_1} \max_{r\in[K]} \mathbb{1}\{E_{i,r} \leq Q_r(\beta_r)\} \ \geq \ \tau,$$

by the feasibility of $\beta$ (5).

*(c) Objective (6a).* By telescoping of increments,

$$\sum_{r=1}^{K}\sum_{\ell=1}^{n_1} c_{r,\ell}\, z_{r,\ell} = \sum_{r=1}^{K}\sum_{\ell=1}^{\ell_r} \left(E_{(\ell),r}^d - E_{(\ell-1),r}^d\right) = \sum_{r=1}^{K} E_{(\ell_r),r}^d = \sum_{r=1}^{K}\left(Q_r(\beta_r)\right)^d.$$

Thus $(z,y)$ is feasible for (6) and preserves the objective.

**(ii) From MILP to quantile problem.** Let $(z,y)$ be feasible for (6). For each $r$, set $\ell_r^\star := \max\{\ell : z_{r,\ell} = 1\}$ (define $\ell_r^\star = 0$ if all $z_{r,\ell} = 0$), $Q_r^\star := E_{(\ell_r^\star),r}$, and pick $\beta_r^\star$ with $Q_r(\beta_r^\star) = Q_r^\star$. Then

$$\max_r \mathbb{1}\{E_{i,r} \leq Q_r^\star\} = \mathbb{1}\left\{\sum_{r,\ell} a_{i,r,\ell} z_{r,\ell} \geq 1\right\} \ \geq \ y_i,$$

so $\sum_i \max_r \mathbb{1}\{E_{i,r} \leq Q_r^\star\} \geq \sum_i y_i \geq \tau$, i.e., $\beta^\star$ is feasible for (5). Moreover,

$$\sum_{r=1}^{K}\left(Q_r(\beta_r^\star)\right)^d = \sum_{r=1}^{K} E_{(\ell_r^\star),r}^d = \sum_{r=1}^{K}\sum_{\ell=1}^{n_1} c_{r,\ell}\, z_{r,\ell},$$

so the objective values coincide.

**(iii) Equivalence.** The two optimization problems are feasible and objective-preserving in both directions, establishing a one-to-one correspondence between solutions of (5) and (6). Hence the MILP is an exact reformulation of the original problem, with identical feasible sets and optimal values. □

## D.2    PROOF OF THEOREM 3.3

*Proof.* Let $\mathcal{D}_{\text{calib}} = \{(X_i, Y_i)\}_{i=1}^{n_2}$ be the calibration set and $(X_{\text{test}}, Y_{\text{test}})$ the test point that is exchangeable with $\mathcal{D}_{\text{calib}}$. Condition on the exploration split $\mathcal{D}_{\text{split}}$ and the optimized threshold vector $T(\alpha)$ from Stage 2, which are fixed before calibration. Define the nonconformity scores

$$S_i = \min_{r\in[K]} \frac{E_{i,r}}{T_r(\alpha)}, \quad i = 1,\ldots,n_2, \qquad S_{\text{test}} = \min_{r\in[K]} \frac{E_{\text{test},r}}{T_r(\alpha)}.$$

The calibration factor $t_\alpha$ is computed:

$$t_\alpha = \min\Big\{\tau \in \mathbb{R} : \frac{1}{n_2 + 1}\sum_{i=1}^{n_2} \mathbb{1}\{S_i \le \tau\} \ge 1 - \alpha\Big\}.$$

By exchangeable assumption, the scores $\{S_1, \ldots, S_{n_2}, S_{\text{test}}\}$ are exchangeable, and the rank of each $S_i$ is uniformly distributed. Therefore

$$\Pr\big(S_{\text{test}} \le t_\alpha\big) \ge 1 - \alpha.$$

Finally, we have the following equivalence

$$
\begin{aligned}
S_{\text{test}} \le t_\alpha &\iff \min_{r \in [K]} \frac{E_{\text{test},r}}{T_r(\alpha)} \le t_\alpha \\
&\iff \exists\, r \in [K] \text{ s.t. } E_{\text{test},r} \le t_\alpha\, T_r(\alpha) \\
&\iff \exists\, r \in [K] \text{ s.t. } \|Y_{\text{test}} - \hat{Y}_{\text{test},\,j(r)}\| \le t_\alpha\, T_r(\alpha) \\
&\iff \exists\, r \in [K] \text{ s.t. } \|Y_{\text{test}} - \hat{Y}_{\text{test},\,j(r)}\| \le R_r(\alpha) \quad (\text{since } R_r(\alpha) := t_\alpha T_r(\alpha)) \qquad (12) \\
&\iff Y_{\text{test}} \in \bigcup_{r=1}^{K}\Big\{y :\ \|y - \hat{Y}_{\text{test},\,j(r)}\| \le R_r(\alpha)\Big\} \\
&\iff Y_{\text{test}} \in \hat{\mathcal{C}}_{\text{ORCA}}(X_{\text{test}}).
\end{aligned}
$$

Thus

$$\Pr\big(Y_{\text{test}} \in \hat{\mathcal{C}}_{\text{ORCA}}(X_{\text{test}})\big) = \Pr\big(S_{\text{test}} \le t_\alpha\big) \ge 1 - \alpha,$$

which completes the proof. $\qquad\square$

### D.3  PROOF OF PROPOSITION 3.4

*Proof.* Let $v_d$ denote the volume of the unit ball in $\mathbb{R}^d$. A classical asymptotic result in the nearest–neighbor estimation literature states that if $m = m_K \to \infty$ with $m/K \to 0$ as $K \to \infty$, then the $j$-th nearest–neighbor radius satisfies

$$R_j(z) = \Big(\frac{j}{K\, p(z|x)\, v_d}\Big)^{1/d}(1 + o(1)),$$

with probability $1 - o(1)$, uniformly over $z \in \mathcal{Y}$ and $1 \le j \le m$ (Loftsgaarden & Quesenberry, 1965; Zhao & Lai, 2022).

Averaging over $j = 1, \ldots, m$ gives

$$\bar{R}_m(z) = \frac{1}{m}\sum_{j=1}^{m} R_j(z) = \big(v_d\, p(z \mid x)\big)^{-1/d} K^{-1/d} \cdot \frac{1}{m}\sum_{j=1}^{m} j^{1/d}\,(1 + o(1)).$$

Since $t \mapsto t^{1/d}$ is increasing, the integral comparison

$$\int_0^m t^{1/d}\, dt \le \sum_{j=1}^{m} j^{1/d} \le 1 + \int_1^m t^{1/d}\, dt$$

implies

$$\sum_{j=1}^{m} j^{1/d} = \frac{d}{d+1}\, m^{1+1/d} + O(1).$$

Therefore,

$$\frac{1}{m}\sum_{j=1}^{m} j^{1/d} = \frac{d}{d+1}\, m^{1/d}\,(1 + o(1)),$$

and hence

$$\bar{R}_m(z) = (v_d)^{-1/d}\, \frac{d}{d+1}\, p(z \mid x)^{-1/d}\, K^{-1/d}\, m^{1/d}\,(1 + o(1)).$$

Thus $\bar{R}_m(z)$ is, up to a constant independent of $z$, proportional to $p(z \mid x)^{-1/d}$. In particular, if $p(z \mid x) > p(z' \mid x)$, then $\bar{R}_m(z) < \bar{R}_m(z')$ with probability $1 - o(1)$, so the averaged nearest-neighbor distance ranking preserves the oracle density ordering. Applying this comparison uniformly to all sample points $\{Z_k\}_{k=1}^K$ shows that the ordering induced by $\bar{R}_m(\cdot)$ coincides with that of $p(\cdot \mid x)$, which completes the proof. $\qquad\square$

## D.4 PROOF OF THEOREM 3.5

*Proof of Theorem 3.5.* Fix $x$ and write $p(\cdot) = p(\cdot \mid x)$. Assume $\mathcal{Y}$ compact, $p$ continuous with $0 < c_- \leq p \leq c_+ < \infty$, and $\mu\{p = \tau^\star(x)\} = 0$. For simplicity, here we assume the samples in exploration and calibration splits are i.i.d., and the generated samples are i.i.d. from $p(\cdot)$.

**Step 1: Empirical Optimization to Population Optimization.** For rank $r \in [K]$, let $F_{r,K}$ be the c.d.f. of $E_{i,r}$ and $\hat{F}_{r,K,n_1}$ its empirical c.d.f. By DKW inequality and a union bound argument,

$$\max_{1 \leq r \leq K} \sup_{\beta \in [\varepsilon, 1-\varepsilon]} \left| Q_r(\beta) - q_r(\beta) \right| = O_{\mathbb{P}}\left(\sqrt{\frac{\log K}{n_1}}\right) \to 0,$$

assuming the positivity of the density of $F_{r,K}$. Define the empirical and population programs

$$\min_{\beta \in [\varepsilon, 1-\varepsilon]^K} J_{K,n_1}(\beta) := \sum_{r=1}^K Q_r(\beta_r)^d \quad \text{s.t.} \quad C_{K,n_1}(\beta) := \frac{1}{n_1} \sum_{i=1}^{n_1} \max_r \mathbf{1}\{E_{i,r} \leq Q_r(\beta_r)\} \geq 1-\alpha,$$

$$\min_{\beta \in [\varepsilon, 1-\varepsilon]^K} J_K(\beta) := \sum_{r=1}^K q_r(\beta_r)^d \quad \text{s.t.} \quad C_K(\beta) := \mathbb{E}\left[\max_r \mathbf{1}\{E_r \leq q_r(\beta_r)\}\right] \geq 1 - \alpha.$$

For the constraint part, we can directly apply the Glivenko–Cantelli theorem and derive

$$\sup_\beta |C_{K,n_1}(\beta) - C_K(\beta)| \to 0.$$

For the objective, we can use the uniform consistency of quantile function, so that

$$\sup_\beta |J_{K,n_1}(\beta) - J_K(\beta)| \to 0 \tag{13}$$

Hence the empirical optimizer $\hat{\beta}_{K,n_1}$ is asymptotically optimal for the population program (argmin consistency). Let

$$\hat{\mathcal{C}}_{K,n_1}(x) = \bigcup_{r=1}^K B\left(Z_{(r)}, Q_r(\hat{\beta}_{K,n_1,r})\right), \qquad \mathcal{C}_K^*(x) = \bigcup_{r=1}^K B\left(Z_{(r)}, q_r(\beta_{K,r}^*)\right),$$

where $\beta_K^*$ is a population optimizer.

**Step 2: Population cutoff and convergence to HDR.** By Proposition 3.4, with probability $1 - o(1)$ the order $r \mapsto Z_{(r)}$ matches decreasing $p$. Consider the population program with objective $\sum_r q_r(\beta_r)^d$ and coverage constraint $C_K(\beta) \geq 1 - \alpha$. The objective is coordinatewise increasing in the radii $\{q_r(\beta_r)\}$, while, for small radius, the mass gain per volume around center $Z_{(r)}$ is proportional to $p(Z_{(r)})$ (first-orde expansion). The threshold $\tau_K$ will be in the form of such that

$$q_r(\beta_{K,r}^*) > 0 \ \Rightarrow \ p(Z_{(r)}) \geq \tau_K, \qquad q_r(\beta_{K,r}^*) = 0 \ \text{otherwise}.$$

The radii of the ball will be vanish with order $O(K^{-1/d})$ Finally, the coverage constraint $C_K(\beta_K^*) = 1 - \alpha$ forces $\tau_K \to \tau^\star(x)$, the unique level with $\int_{L_{\tau^\star}} p = 1 - \alpha$, hence

$$\mu(\mathcal{C}_K^*(x) \triangle L_{\tau^\star}(x)) \to 0.$$

**Step 3: Calibration.** Split-conformal scales radii by $t_\alpha$. Since the pre-calibrated set already attains asymptotic coverage $1 - \alpha$ and the score distribution concetrates at $1$, $t_\alpha \to 1$ in probability, so

$$\mu(\mathcal{C}_{K,n_1,n_2}(x) \triangle L_{\tau^\star}(x)) \xrightarrow{\mathbb{P}} 0.$$

$\qquad\square$

Table 3: Summary of real-world datasets.

| Dataset | # Features | Response Dimension | # Samples |
|---|---|---|---|
| Blog Feedback (Blog) (Buza, 2014a) | 280 | 1 | 52,397 |
| Facebook Comment Volume (Fb1) (Singh, 2015a) | 53 | 1 | 40,948 |
| Facebook Comment Volume (Fb2) | 53 | 1 | 81,311 |
| MEPS (Year 19)[2] | 139 | 1 | 15,785 |
| MEPS (Year 20) | 139 | 1 | 17,541 |
| MEPS (Year 21) | 139 | 1 | 15,656 |
| Protein Structure (Bio) (Rana, 2013) | 9 | 1 | 45,730 |
| Temperature (tem, 2020) | 21 | 1 | 7,590 |
| Bike[3] | 11 | 2 | 735,502 |

# E    IMPLEMENTATION DETAILS

## E.1    MODEL TRAINING AND HYPERPARAMETER SETTINGS

For conditional distribution estimation in the 1-dimensional case, we implemented quantile regression forests (QRF) (Meinshausen, 2006) with 100 trees, a minimum of 50 samples per node split, and a quantile grid ranging from 1% to 99%. The trained model serves as the base model for CQR, CHR, DCP, DistSplit, CRD, PCP, and ORCA. The implementation is based on the publicly available repository.[1] For methods requiring explicit density estimation, such as CDsplit, we employed a mixture density network (MDN) (Bishop, 1994) with 10 Gaussian components, dropout rate of 0.2, hidden layer size of $(100, 100)$, and 500 training epochs. The implementation is based on the Python package `cde` (Rothfuss et al., 2019). For R2CCP, we used the default configuration from the author's public repository and trained for 500 epochs. In the multivariate setting, we applied Bonferroni correction to construct coverage intervals for CHR, CQR, DCP, DistSplit, CDsplit, and R2CCP, using the same base models as in the 1-dimensional case. For CRD, PCP, and ORCA, we used MDN to model multivariate responses.

## E.2    REAL DATASETS

We evaluate the performance of ORCA and baseline methods on a diverse collection of real-world datasets spanning a range of application domains. These datasets vary in input dimensionality, response dimension, and sample size, and include both univariate and multivariate prediction tasks. A summary of the datasets used in our experiments is provided in Table 3.

# F    EXPERIMENTS RESULTS

In this section, we present the additional experimental results for synthetic data, and real data.

## F.1    SYNTHETIC DATA RESULTS

We present the visualizations of the prediction set provided by different baseline methods on S-shape, Circles, Spirals and Unbalanced clusters in Figure 4. It is clear that our method can consistently provide discontinuous and adaptive prediction set, resulting in high efficiency. We also investigate the effect of $K$ on coverage efficiency. We provide an example in Figure 5, which presents the ratio of PCP prediction set length over ORCA prediction set length for S-shape data under different choices of $K = 3, 5, 10, 15, 20$ and different numbers of calibration data. The curve plot represents the mean, and the shaded area indicates the standard error of the ratio over 100 experiments. Under all settings, the ratio consistently remains above 1, showing that ORCA provides a more efficient prediction set. Additionally, the ratio decreases as $K$ increases, suggesting that the benefit of ORCA decreases with larger $K$.

---

[1] https://github.com/msesia/chr

[2] https://meps.ahrq.gov/mepsweb/

[3] https://citibikenyc.com/system-data

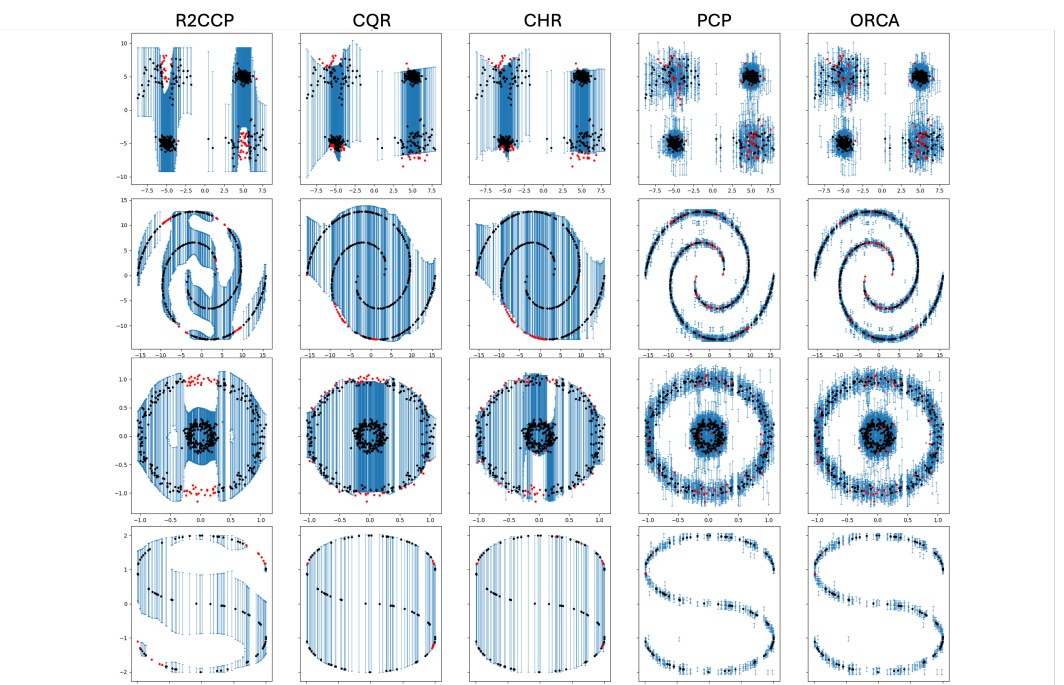

Figure 4: Comparison of prediction set ($\alpha = 0.1$) generated by different methods on 1-dimensional synthetic data: S-shape, Circles, Spirals, and Unbalanced clusters (from top to bottom row). Blues lines: the predictive intervals of each method; Black dots: test points that are covered by the predictive sets; Reds dots: test points not covered. We plot the figure with 100, 500, 400, and 500 random samples for each data, respectively.

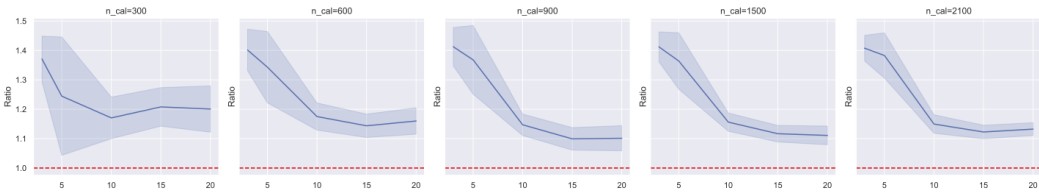

Figure 5: Ratio of PCP prediction set length over ORCA prediction set length under different $K = 3, 5, 10, 15, 20$ and number of calibration data for S-shape data. The blue curve represents the mean, and the shaded area indicates the standard error of the ratio. We observe that the ratio remains consistently above 1, demonstrating that ORCA achieves better efficiency.

