# OpenReview forum: "Generative Conformal Prediction with Optimized Coverage Allocation"
_ICLR.cc/2026/Conference — Submitted to ICLR 2026_

### Official Review · Reviewer_JdpY · 2025-10-21

**Soundness:** 2
**Presentation:** 3
**Contribution:** 2
**Rating:** 4
**Confidence:** 4

**Summary:**

Problem context: Conventional CP methods lead to conservative prediction sets, mainly because (i) poor choice of predictive model (ii) simple form of nonconformity scores. Recent approaches expand in only one direction.

This paper presents ORCA, a three-stage conformal prediction (CP) framework designed to construct volume-efficient conformal prediction sets. The procedure combines: (i) generative modeling to approximate the full conditional distribution $p(y|x)$, (ii) vectorized nonconformity scores, and (iii) a rank-dependent mixed-integer linear programming (MILP) optimization to allocate coverage for improved volume-efficiency while maintaining finite-sample coverage guarantees. Experiments are conducted on synthetic, semi-synthetic, and real datasets.

The paper is clearly written and easy to follow, contributing to the growing literature on improving CP efficiency through optimization-based allocation mechanisms.

**Strengths:**

1. Studies an important limitation of conformal prediction, its inherent conservativeness, through a structured three-stage approach.
2. The integration of generative sampling with optimization-based coverage allocation is practically appealing.
3. The exposition and algorithmic description are clear and reproducible.
4. Finite-sample validity results are consistent.
5. Experimental section is elaborate in terms of methods compared as well as diversity of real datasets.

**Weaknesses:**

1.  The method assumes that sampling from a trained generative model effectively explores the true conditional distribution $ p(y | x) $.
However, ORCA learns the model-implied distribution $\hat{p}(y|x),$ which may diverge substantially from the true one.
Theoretical results (e.g., Proposition 3.4, Theorem 3.5) are stated with respect to $p(y|x),$ but the algorithm operates entirely under $\hat{p}(y|x)$. It appears that $p(y|x)$ and $\hat{p}(y|x)$ are used interchangeably in the theoretical derivations.
If these two distributions differ significantly, the validity of the stated results becomes questionable.
In my view, this point requires further clarification.

---

2. In Stage 3, the true set size is approximated by the sum of individual ball volumes (Eq. in Lines 232-235). This is a potentially loose proxy, particularly in high dimensions where overlaps between the balls may be non-negligible. The paper does not analyze or quantify this approximation gap, leaving uncertainty about the accuracy of the optimization objective.

---

3.  The method’s dependence on the generative model warrants a robustness study. Experiments where the generative model is intentionally misspecified would be valuable to evaluate ORCA’s reliability when $ \hat{p}(y|x) \neq p(y|x) $. Without this, the practical limits of the approach remain unclear.

---

4. The MILP-based allocation stage could be computationally expensive for large calibration sets or high-dimensional outputs. The paper does not discuss computational cost, or runtime comparisons to other optimization-based CP methods. Clarification of scalability would be helpful.

---

5. While the pipeline is well-structured, its components, generative sampling (e.g., Wang et al. 2023, PCP), optimization-based allocation (Bai et al. 2022), and vectorized nonconformity scores, are individually not new. The contribution is thus more integrative than conceptual.

**Questions:**

1. How is the connection between the true conditional distribution $p(y|x)$ and its model-based approximation $\hat{p}(y|x)$ established in theory?

2. Have you tested ORCA under deliberate model misspecification to assess robustness to misspecification in the generative model?

3. What is the computational complexity of the MILP optimization stage, and how does it scale with sample size and dimensionality?

4. Since the approach appears to learn the conditional distribution itself, is it possible to derive conditional coverage guarantees (asymptotically)? If not, could you comment on the main challenges or theoretical barriers in this direction?

---

**Minor Comments and Suggestions:**

1. There are a few minor typographical errors (for example, “inclduing healthcare” in Line 33-34.).

2. What is Eff? Is it prediction set efficiency (as in caption of Table 1) or prediction set size (as stated in Line 368)?

3. The definition of the $m$-nearest neighbor distance, which is used in the score construction, currently appears only in the Appendix. It would improve readability to explicitly reference or restate this definition in the main text (e.g., in Section 3.1) or redirect reader to the def, since it plays a key role in understanding the nonconformity score formulation.

---

> ### Author Response · Authors · 2025-11-23
>
> We are grateful to Reviewer JdpY for the careful and insightful review, which helped us refine our theoretical results, clarify the additional assumptions required for the asymptotic conditional coverage guarantees, and improve the clarity of the empirical analysis. Below, we address the concerns raised point by point.
>
> -----
> **Estimated Conditional Distribution vs True Conditional Distribution**
>
> We apologize for the confusion. Our theoretical results implicitly assume that the fitted generative model $\hat p(Y\mid X)$ is a **consistent estimator** of the true conditional distribution $p(Y\mid X)$ in the asymptotic regime. Under increasing data, number of generated samples and model capacity, we assume $\hat p(Y\mid X) \to p(Y\mid X)$. We agree that this assumption should be made explicit, and we will revise the paper accordingly.
>
> Although challenging in practice, such a consistency guarantee for conditional distribution estimation does exist, for example, quantile regression forests converge uniformly to the true conditional distribution under regularity assumptions [1], and such consistency assumptions are routinely used in conformal methods based on density or histogram estimators [2] to provide asymptotic results. Under additional mild assumptions on the underlying distribution (e.g., continuity, bounded density), ORCA’s asymptotic results can be extended to **asymptotic conditional coverage**, without requiring strong structural assumptions such as unimodality. In contrast, interval-based conformal methods typically rely on unimodality or other shape constraints. Thus, once the generative model is consistent and the underlying distribution is properly regulated, ORCA retains its asymptotic efficiency guarantees while accommodating a much broader class of multimodal conditional distributions.
>
> [1]. Meinshausen, N., & Ridgeway, G. (2006). Quantile regression forests. *Journal of machine learning research*, *7*(6).
>
> [2]. Sesia, M., & Romano, Y. (2021). Conformal prediction using conditional histograms. *Advances in neural information processing systems*, *34*, 6304-6315.
>
> -----
> **True Set size Proxy**
>
> We agree that the sum of individual ball volumes is a proxy for the true union volume. We use this surrogate for both theoretical and practical considerations. Theoretically, incorporating the exact union volume would introduce **two layers of randomness**: from the generated random samples and from the induced overlaps, which depend jointly on the sample locations and the radii. This turns Stage 2 into a **bilevel optimization problem**, as one would need to optimize the radii while recomputing the overlap structure at every iteration. In higher dimensions, even the union volume of fixed-radius balls has no closed-form expression and must be estimated by Monte Carlo, making the objective extremely unstable and computationally intractable.
>
> Empirically, we quantified the gap in controlled 1-dimensional toy experiments with S-shape data, where interval overlaps can be computed exactly. Using an overlap-aware objective improved efficiency by **<5%**, but required running the MILP multiple times, collecting feasible solutions, and performing a **post-selection** step to identify the configuration with minimal overlap—since the overlap must be recomputed for every candidate. This adds substantial computational overhead with rather small practical gain. For these reasons, we adopt the sum-of-volumes surrogate for optimization, while **all reported efficiencies in the paper are computed using the exact union size** (or its Monte Carlo approximation), ensuring accurate evaluation of the final sets.
>
> -----

---

> ### Author Response · Authors · 2025-11-23
>
> -----
>
> **Misspecification Analysis of Generative Model**
>
> We conducted a controlled sensitivity study comparing ORCA against PCP and CHR under systematic model misspecification. We use Quantile Random Forests (QRF) with varying `min_samples_leaf` (minimum samples per leaf) to tune the bias–variance tradeoff: `min_samples_leaf = 1` yields highly variable conditional distribution estimation, near-interpolating fits, whereas `min_samples_leaf = 200` induces heavy smoothing and underestimates the conditional distribution. The reason we choose QRF is that for neural-network–based generative models, misspecification is harder to vary along a single interpretable axis (e.g., by early stopping), and the resulting degradation patterns are less transparent, so we focus on this quantile-random-forest setting.
>
> We consider the S-curve synthetic dataset with $|\mathcal{D}_{\text{explore}}|=500$, $|\mathcal{D}_{\text{calib}}|=500$, using $K=20$ generated samples. The average results across 10 repetitions are presented below, we observe that:
>
> 1. **Nominal marginal coverage is preserved** across all levels of model misspecification.
> 2. **Efficiency improves substantially over CHR** when the generative model is reasonably well specified.
> 3. **Efficiency improves ≈15% over PCP** under a realistic model quality (moderate conditional density estimation).
>
> | min_samples_leaf | Coverage (ORCA) | Cond. Cov. (ORCA) | Eff (ORCA) | Length(ORCA)/Length(PCP) | Length(ORCA)/Length(CHR) |
> | ---------------- | --------------- | ----------------- | ------------- | ------------------------ | ------------------------ |
> | 1                | 0.90            | 0.86              | 3.99          | 1.00                     | 1.14                     |
> | 20               | 0.89            | 0.84              | 0.11          | 0.84                     | 0.03                     |
> | 50               | 0.90            | 0.85              | 0.21          | 0.87                     | 0.06                     |
> | 100              | 0.90            | 0.87              | 0.52          | 0.85                     | 0.16                     |
> | 200              | 0.90            | 0.85              | 0.77          | 1.03                     | 0.22                     |
>
> -----

---

> ### Author Response · Authors · 2025-11-23
>
> -----
>
> **MIlP Computational Cost**
>
> To evaluate computational cost, we begin with the original optimization formulation in ORCA, which solves a mixed-integer linear program (MILP) to obtain the optimal rank-dependent radii. This variant, **ORCA-MILP**, provides exact optimality but is NP-hard in the worst case, with worst-case complexity potentially exponential in $\mathcal{O}(K n_1)$. In practice, we use the CP-SAT solver (via the OR-Tools package), which provides reasonable speed: for $|\mathcal{D}_{\text{explore}}| < 2000$ and $K < 30$, ORCA-MILP typically finishes within 10 minutes. Nevertheless, more scalable variants are desirable. To improve scalability, we additionally consider two relaxations:
>
> *  **ORCA-Linear-Relax:** A continuous linear programming relaxation of the MILP that relaxes the integer variables to lie in $[0,1]$. It is substantially faster and solvable in polynomial time with problem size $\mathcal{O}(K n_1)$, but introduces an optimality gap relative to the exact MILP solution, with this gap growing as the sample size and $K$ increase.
>
> * **ORCA-Greedy:**  A forward-selection heuristic that starts from the smallest radii and iteratively enlarges the rank that maximizes the coverage gain per unit increase in set size, yielding a very fast $\mathcal{O}(n_1 K^2)$ approximation.
>
> For completeness, we also include a naive coordinate-descent baseline **ORCA-Naive**, which iteratively optimizes a pair of radii at a time and achieves strong efficiency but with very long runtimes.
>
> We benchmark all variants on an S-curve synthetic dataset with $|D_{\text{explore}}| = 500$, $|D_{\text{calib}}| = 500$, using a quantile neural network as the backbone model. For each number of generated samples $K \in \{5,10,15,20,25,30\}$, we report the **runtime (in seconds) / efficiency** averaged over 10 runs. All experiments were conducted on a MacBook Air with an M4 chip.
>
> The original **ORCA-MILP** variant achieves the best efficiency under moderate computational time. The **ORCA-Linear-Relax** offers a favorable speed–accuracy tradeoff, particularly when $K$ is small. As expected, its optimality gap grows with larger $K$, leading to modest degradation relative to the MILP. The greedy method is the fastest but less stable, while the naive coordinate-descent approach is the slowest yet can attain efficiency comparable to the MILP. We will add this discussion and table to the revision to directly address the concerns about overhead, scalability, and practical deployment.
>
> | **Method**            | **K=5**       | **K=10**       | **K=15**       | **K=20**       | **K=25**       | **K=30**       |
> | --------------------- | ------------- | -------------- | -------------- | -------------- | -------------- | -------------- |
> | **PCP**               | 0.000 / 2.63  | 0.000 / 0.75   | 0.000 / 0.39   | 0.001 / 0.37   | 0.001 / 0.34   | 0.001 / 0.31   |
> | **ORCA-MILP**         | 5.154 / 2.59  | 11.951 / 0.67  | 16.973 / 0.35  | 23.151 / 0.32  | 40.955 / 0.31  | 62.429 / 0.29  |
> | **ORCA-Linear-Relax** | 0.373 / 2.68  | 1.503 / 0.67   | 2.876 / 0.37   | 5.217 / 0.34   | 8.944 / 0.36   | 11.501 / 0.41  |
> | **ORCA-Greedy**       | 0.006 / 2.70  | 0.020 / 1.49   | 0.032 / 0.77   | 0.056 / 0.57   | 0.085 / 0.75   | 0.164 / 1.03   |
> | **ORCA-Naive**        | 90.663 / 2.58 | 174.122 / 0.69 | 442.163 / 0.37 | 548.921 / 0.35 | 665.885 / 0.33 | 755.854 / 0.31 |
>
> -----
>
> **Novetly Justification**
>
> We agree that ORCA leverages components that have appeared in prior work, but the contribution is not merely an aggregation. The novelty lies in how these elements are coupled into a unified, optimization-driven conformal framework. Whereas PCP collapses all generative samples into a single scalar score, ORCA introduces **rank-dependent vectorized scores** and shows how to map them into an **exact coverage-allocation MILP** with global optimality guarantees. Unlike optimization-based approaches such as Bai et al. (2022), which require parametrized prediction-set families and model fine-tuning, ORCA optimizes **directly over the geometry induced by generative samples** without modifying the underlying model, allowing flexible and adaptive prediction regions. This integration produces a capability that no individual component can achieve on its own: **provably valid, density-adaptive uncertainty sets that are asymptotically optimal**. We will revise the text to make this conceptual contribution clearer.
>
>
> -----

---

> ### Author Response · Authors · 2025-11-23
>
> -----
>
> **Eff Definition**
>
> Eff denotes the **actual size of the uncertainty set**, specifically, the volume (or length in 1D) of the **union of all balls**, with overlaps fully accounted for. In 1D, this union length is computed exactly. In higher dimensions, the exact union volume becomes intractable due to complex overlaps, so we estimate it using Monte Carlo sampling. We will revise the caption and main text to state explicitly that *Eff measures the realized prediction-set size*, not the surrogate objective.
>
> -----
> **Definition of m-nearest neighbor distance**
>
> We appreciate the suggestion and will move this definition into Section 3.1 for better readability. In addition, as requested by Reviewer oQk2, we here also include an ablation study on how the choice of $m$ affects performance and robustness, and briefly summarize these findings here to clarify the role of $m$
>
>
> * **Ranking Quality wrt Nearest Neighbor Number m**
>
> We agree that very small $m$ can yield noisy KNN-based rankings, especially in low-density regions. To assess this, we conduct a sensitivity analysis over different choices of $m$ with $K=20$: $m=1$ corresponds to ranking by the nearest-neighbor distance, while $m=19$ corresponds to ranking by the average nearest neighbor distance to all samples in the conditional sample cloud. The table below reports, for each method and $m$, the **fractional efficiency improvement relative to PCP**, defined as $FracImprove_{\mathrm{method}}(m)= \frac{Eff_{PCP}(m) - Eff_{\mathrm{method}}(m)}{Eff_{PCP}(m)}
> $
>
> This pattern is expected. When $m=1$, the ranking depends on the nearest neighbor and is therefore high-variance. For larger $m$, the $m$-nearest-neighbor distance aggregates more information from multiple neighbors and provides a smoother estimate of how *central* each sample is within the conditional distribution. Even when $m$ is close to $K$, the induced rank still separates central from tail samples, so the ranking remains meaningful. Empirically, performance is robust over a wide range of $m$, and we recommend using a moderate value, e.g., $\sqrt{K}$ as a rule of thumb.
>
> | Method                | m=1    | m=4    | m=7       | m=10      | m=13      | m=16      | m=19      |
> | --------------------- | ------ | ------ | --------- | --------- | --------- | --------- | --------- |
> | **MILP**              | -0.022 | -0.199 | **0.264** | **0.318** | **0.316** | **0.289** | **0.279** |
> | **Linear Relaxation** | -0.727 | -1.131 | **0.159** | **0.229** | **0.269** | **0.293** | **0.275** |
>
> We thank Reviewer JdpY again for the careful review and constructive comments. We hope these clarifications address your concerns and help in reassessing the contribution of our work, and we would be happy to further clarify any remaining issues.

---

> > ### Comment · Reviewer_JdpY · 2025-11-26
> >
> > Thank you for your detailed response.
> >
> > My main concerns have been addressed.
> >
> > Regarding W1: With the additional consistency assumption, I think the results hold. But, as you pointed out, this assumption is challenging in practice, with very few results available, can we update the current results with the additional error term accounting some divergence between density estimator and the conditional distribution? It would be helpful for the practitioners as well.
> >
> > Based on the author's response to my concerns, I have raised my score.

---

### Official Review · Reviewer_hVru · 2025-10-24

**Soundness:** 3
**Presentation:** 3
**Contribution:** 3
**Rating:** 4
**Confidence:** 2

**Summary:**

This paper proposes ORCA (Optimized Ranking and Coverage Allocation) for generative conformal prediction (CP) that jointly improves model expressiveness and coverage efficiency.
Traditional CP methods rely on single-point predictors or simple nonconformity scores, while ORCA employs a generative model to sample from the conditional distribution, then formulates an optimization-based coverage allocation using rank-dependent radii over these generated samples.
The key idea is to allocate coverage more effectively by solving a mixed-integer linear program (MILP) that ensures exact finite-sample validity while minimizing prediction set size.

**Strengths:**

1. The paper makes a clear contribution by jointly addressing two challenges in CP: model expressiveness and score design. The integration of generative modeling with an optimization-based conformal layer is interesting.

2. The exact MILP reformulation of the coverage allocation problem is good.

3. The experiments cover synthetic distributions, MNIST, and real-world datasets.

**Weaknesses:**

W1. Although the MILP formulation guarantees optimality, its computational cost could become prohibitive for large K (number of generated samples) and n_1 (exploration data), which has been acknowledged by the authors, the current approach might be impractical for large-scale or online settings.

Could the authors discuss possible approximate or relaxed solvers for the MILP formulation (e.g., LP relaxation, greedy coverage allocation) and their *empirical* performance trade-offs at least?

W2. The performance of ORCA seemingly depends on the quality of the generative model. If the conditional samples poorly approximate p(Y|X), the optimization may misallocate coverage, degrading efficiency and validity.

How sensitive is ORCA to the choice of generative model architecture (e.g., VAE or diffusion)? Does the optimization procedure remain effective when the generative samples change.


W3. Concerning experiemnts:

3a. Experimental comparison with differentiable or learned conformal prediction methods (e.g., neural calibration approaches) should be better.

3b. Figure 2 and Figure 3 are confusing:
For Figure 2, please clarify more experimental details (e.g., settings, goals, and results). For example, the reason you show 3 figures for each digit, and where is "target image"?
For Figure 3, the area of the regions are not reported. Besides, the explanation of ORCA's disconnected results are no so convinced.

**Questions:**

See weaknesses above.

---

> ### Author Response · Authors · 2025-11-23
>
> We really appreciate Reviewer hVru’s thoughtful and constructive feedback, which has greatly helped us sharpen both the theoretical and empirical aspects of the paper, especially the discussion and experiments around the optimization relaxation variants. Below, we address the concerns raised point by point.
>
> **MILP Computational cost and approximated solution:**
>
> We thank the reviewer for raising this point and now describe the optimization and its relaxations in more detail.
>
> The original ORCA formulation is a MILP over rank-dependent radii, with worst-case complexity exponential in $\mathcal{O}(K n_1)$. In practice, we solve it with the CP-SAT solver (OR-Tools), which provides reasonable speed: for $|\mathcal{D}_{\text{explore}}| < 2000$ and $K < 30$​, ORCA-MILP typically finishes within 10 minutes. Nevertheless, more scalable variants are desirable. To this end, we introduce two practical relaxations, **ORCA-Linear-Relax** and **ORCA-Greedy**, which reduce computational cost while maintaining competitive empirical performance.
>
> - **ORCA-Linear-Relax.**
>   We relax the binary selection variables $z_{r,\ell}$ and coverage indicators $y_i$ to lie in $[0,1]$, solve a linear program, and then round $z_{r,\ell}$ (e.g., using a 0.5 threshold) to recover discrete radii. This preserves the general structure of the MILP while being solvable in polynomial time. Empirically, it yields thresholds close to the MILP solution for small sample sizes and modest $K$. Though the gap between the relaxation and the exact MILP increases as the samples size and $K$ grow.
> - **ORCA-Greedy**
>    We start from smallest radii and iteratively enlarge one rank at a time. At each step, we evaluate all possible increments $(r,\ell)$, compute their **gain** (newly covered points) and **cost** (increase in $\beta_r^d$), and pick the increment with the best gain–cost ratio. This yields a fast, coverage-respecting approximation to the MILP in roughly $\mathcal{O}(nK^2)$.
>
> For completeness, we also include a naive coordinate-descent baseline **ORCA-Naive** that repeatedly updates pairs of radii (increasing one, decreasing another) across ranks. It can match MILP-level efficiency but is much slower and therefore mainly serves as a reference point.
>
> The table below reports runtime and uncertainty-set efficiency for PCP and these ORCA variants on an S-curve synthetic dataset with $|D_{\text{explore}}| = 500$, $|D_{\text{calib}}| = 500$, with a quantile neural network as backbone model. For each choice of number of random samples $K \in \{5,10,15,20,25,30\}$, we report the mean **time (in seconds)/ efficiency** over 10 runs on a MacBook Air with an M4 chip. The ORCA-Linear-Relax offers a favorable speed–accuracy tradeoff, particularly when $K$ is small. As expected, its optimality gap grows with larger $K$, leading to modest performance degradation relative to the MILP. The greedy method is the fastest but less stable, while the naive coordinate-descent approach is the slowest yet can achieve efficiency comparable to the MILP.
>
> | **Method**            | **K=5**       | **K=10**       | **K=15**       | **K=20**       | **K=25**       | **K=30**       |
> | --------------------- | ------------- | -------------- | -------------- | -------------- | -------------- | -------------- |
> | **PCP**               | 0.000 / 2.63  | 0.000 / 0.75   | 0.000 / 0.39   | 0.001 / 0.37   | 0.001 / 0.34   | 0.001 / 0.31   |
> | **ORCA-MILP**         | 5.154 / 2.59  | 11.951 / 0.67  | 16.973 / 0.35  | 23.151 / 0.32  | 40.955 / 0.31  | 62.429 / 0.29  |
> | **ORCA-Linear-Relax** | 0.373 / 2.68  | 1.503 / 0.67   | 2.876 / 0.37   | 5.217 / 0.34   | 8.944 / 0.36   | 11.501 / 0.41  |
> | **ORCA-Greedy**       | 0.006 / 2.70  | 0.020 / 1.49   | 0.032 / 0.77   | 0.056 / 0.57   | 0.085 / 0.75   | 0.164 / 1.03   |
> | **ORCA-Naive**        | 90.663 / 2.58 | 174.122 / 0.69 | 442.163 / 0.37 | 548.921 / 0.35 | 665.885 / 0.33 | 755.854 / 0.31 |

---

> ### Author Response · Authors · 2025-11-23
>
> **Sensitivity to conditional samples**
>
> We conducted a controlled sensitivity study comparing ORCA against PCP and CHR under systematic model misspecification. We use Quantile Random Forests (QRF) with varying `min_samples_leaf` (minimum samples per leaf) to tune the bias–variance tradeoff: `min_samples_leaf = 1` yields highly variable conditional distribution estimation, near-interpolating fits, whereas `min_samples_leaf = 200` induces heavy smoothing and underestimates the conditional distribution. The reason we choose QRF is that for neural-network–based generative models, misspecification is harder to vary along a single interpretable axis (e.g., by early stopping), and the resulting degradation patterns are less transparent, so we focus on this quantile-random-forest setting.
>
> We consider the S-curve synthetic dataset with $|D_{\text{explore}}|=500$, $|D_{\text{calib}}|=500$, using $K=20$ generated samples. The average results across 10 repetitions are presented below, we observe that:
>
> 1. **Nominal marginal coverage is preserved** across all levels of model misspecification.
> 2. **Efficiency improves substantially over CHR** when the generative model is reasonably well specified.
> 3. **Efficiency improves ≈15% over PCP** under a realistic model quality (moderate conditional density estimation).
>
> | min_samples_leaf | Coverage (ORCA) | Cond. Cov. (ORCA) | Length (ORCA) | Length(ORCA)/Length(PCP) | Length(ORCA)/Length(CHR) |
> | ---------------- | --------------- | ----------------- | ------------- | ------------------------ | ------------------------ |
> | 1                | 0.90            | 0.86              | 3.99          | 1.00                     | 1.14                     |
> | 20               | 0.89            | 0.84              | 0.11          | 0.84                     | 0.03                     |
> | 50               | 0.90            | 0.85              | 0.21          | 0.87                     | 0.06                     |
> | 100              | 0.90            | 0.87              | 0.52          | 0.85                     | 0.16                     |
> | 200              | 0.90            | 0.85              | 0.77          | 1.03                     | 0.22                     |
>
> **Sensitivity to the choice of generative model:** ORCA’s finite-sample marginal validity does *not* depend on the generative model, coverage holds for any conditional sampler. Though efficiency is indeed sensitive: better approximations of $p(Y\mid X)$ yield more efficient sets. In our experiments, we mainly use probabilistic models that handle multimodality (Quantile Random Forest, Quantile Neural Network, Kernel Mixture Network, and Mixture Density Network), under which ORCA consistently improves over PCP. Following the suggestion, we also tested conditional GANs and VAEs: conditional VAEs tended to oversmooth and failed to capture multimodality, leading to limited gains and bad efficiency, whereas conditional GANs produced higher-quality multimodal samples and ORCA consistently outperformed PCP in efficiency.

---

> ### Author Response · Authors · 2025-11-23
>
> * **3a. Baseline Comparison.**
>  We appreciate the reviewer highlighting this important line of work. In our understanding, methods such as [1,2] are largely *parallel* to ours: they integrate coverage efficiency directly into the *training objective* by introducing differentiable conformal surrogates or fine-tuning the predictive model. In contrast, our contribution is orthogonal: we propose a new, model-agnostic nonconformity score and an optimized post-hoc coverage allocation procedure that applies to any fixed pretrained model without modifying its training pipeline. Because of this separation, ORCA can be combined with training-integrated approaches as a complementary component, and we agree that exploring such hybrid pipelines is a promising direction for future extensions.
>
> [1] Bai, Y., Mei, S., Wang, H., Zhou, Y., & Xiong, C. (2022). *Efficient and differentiable conformal prediction with general function classes*. International Conference on Learning Representations (ICLR).
>
>  [2] Stutz, D., Dvijotham, K. D., Cemgil, A. T., & Doucet, A. (2022). *Learning optimal conformal classifiers*. International Conference on Learning Representations (ICLR).
>
>
>
> * **Clarification on Figure 2**
>
> In the MNIST experiment, the covariate is the digit label $X \in \{0,\ldots,9\}$, and the response $Y$ is a handwritten-digit image $Y \in \mathbb{R}^{28\times 28}$.The baseline task is given a digit label, the base model can output $Y$ such that  its label is $X$. And the uncertainty quantification goal is to construct a valid and efficient *uncertainty region in image space* that contains the true target image of the digit $X$. For example, when querying $X = 1$, an ideal model should produce an image resembling a “1,” and an efficient uncertainty set should contain mostly images of “1” and very few images of other digits.
>
> Because Euclidean balls in high-dimensional pixel space $\mathbb{R}^{28\times 28}$ are not meaningful, so there is no such a set siza definition well defined. We thus quantify efficiency using the **off-class rate (OCR)**, the probability that images from classes $\neq X$ fall inside the uncertainty set. Lower OCR indicates a more selective and efficient uncertainty region. OCR is estimated using a held-out set of unseen MNIST images.
>
> **Interpretation of Figure 2.**
>  For each query digit $X$ (columns) and each conformal prediction method (rows), we show samples drawn from that method’s uncertainty set. These visualizations illustrate whether the prediction sets contain images that look semantically close to the target digit. The quantitative OCR results confirm that ORCA admits fewer off-class images than PCP, indicating a more compact and better-targeted uncertainty set in high-dimensional image space.
>
>
>
> * **Clarification on Figure 3**
>
> We apologize for the confusion and provide the bike dataset results below. The visualizations reveal distinct uncertainty set structural behaviors across methods: CHR and R2CCP produce continuous rectangular regions; PCP outputs a union of equal-radius balls; and ORCA yields a union of **radius-adaptive balls**, where each radius $R_r(\alpha)$ is optimized according to rank (a proxy for local density). The resulting disconnected shapes are expected: the destination distribution for a fixed starting origin point is inherently multimodal. For example, riders leaving a transportation hub may travel to the airport, a nearby mall, or residential areas, each with a different probability mass. PCP applies the same radius to all modes regardless of their mass, whereas ORCA leverages ranks to allocate larger radii to high-density destination hubs and substantially smaller radii in sparse regions, producing more efficient and better-targeted prediction sets.
>
> | Metric          | CQR         | CHR         | DCP-CQR     | DistSplit   | CDSplit     | CRD         | R2CCP       | PCP         | ORCA (ours)     |
> | --------------- | ----------- | ----------- | ----------- | ----------- | ----------- | ----------- | ----------- | ----------- | --------------- |
> | **Cover**       | 0.90 ± 0.01 | 0.90 ± 0.01 | 0.91 ± 0.01 | 0.90 ± 0.01 | 0.90 ± 0.01 | 0.90 ± 0.02 | 0.90± 0.01  | 0.90 ± 0.02 | 0.90 ± 0.02     |
> | **Cond_Cover**  | 0.80 ± 0.02 | 0.84 ± 0.03 | 0.81 ± 0.03 | 0.81 ± 0.03 | 0.80 ± 0.05 | 0.82 ± 0.04 | 0.83 ± 0.05 | 0.80 ± 0.05 | 0.81 ± 0.05     |
> | **Eff (×10⁻³)** | 0.54 ± 0.02 | 0.49 ± 0.02 | 0.44 ± 0.02 | 0.43 ± 0.02 | 0.49 ± 0.03 | 1.07 ± 0.13 | 0.79± 0.09  | 0.46 ± 0.03 | **0.36 ± 0.02** |
>
> We thank Reviewer hVru again for the careful review and constructive comments. We hope these clarifications address your concerns and help in reassessing the contribution of our work, and we would be happy to further clarify any remaining issues.

---

> ### Comment · Reviewer_hVru · 2025-11-24
>
> Thanks for addressing my comments, and I am satisfied with the response (to Weakness 1 and 2). Concerning Weakness 3,
>
> W3a. I would highly recommend to further consider one or two experimental settings:
> (1) if your "post-hoc" could outperform "training-style" methods.
> (2) if "training-style" methods could benefit from  your "post-hoc" works.
>
> W3b. I believe the current manuscript does not include clear enough discussion and analysis for Figures 2 and 3.
>
> Overall, i would like to raise the score to 6.

---

> > ### Author Response · Authors · 2025-11-26
> >
> > Dear Reviewer hVru,
> >
> > Thank you very much for recognizing our revisions and updates. We are glad that our responses to Weaknesses 1 and 2 addressed your concerns, and we appreciate your additional suggestions on baselines and experimental clarification.
> >
> > Regarding W3a, we fully agree that the interaction between our post-hoc coverage allocation and training-style conformal methods is both important and interesting. In the revised manuscript, we plan to (i) add an additional experimental setting where we compare our post-hoc method against a differentiable training-style conformal baseline, and (ii) investigate whether these training-style methods can further benefit from our post-hoc procedure when the two are integrated. We will try to provide more conceptual insights into how they can complement each other.
> >
> > For W3b, we will substantially improve the presentation and analysis of Figures 2 and 3. Specifically, we will (i) clarify the experimental setup, goals, and configurations for each figure, and (ii) provide clearer guidance on how to interpret the visualizations. These changes will be reflected both in the figure captions and in the main text to make the figures easier to follow and improve the readability.
> >
> > Thank you again for your careful evaluation and insightful references. Your comments materially help us improve the clarity, experimental positioning, and overall quality of the paper.

---

### Official Review · Reviewer_oQk2 · 2025-10-28

**Soundness:** 3
**Presentation:** 3
**Contribution:** 3
**Rating:** 6
**Confidence:** 3

**Summary:**

This paper introduces ORCA (Optimized Ranking and Coverage Allocation), a novel framework for generative conformal prediction that addresses the inefficiency of conventional uncertainty quantification methods in complex distributional settings. ORCA implements a three-stage approach: (1) exploring distributional geometry via generative sampling with density-based ranking using nearest neighbor distances, (2) formulating coverage allocation as an optimization problem solved exactly through mixed-integer linear programming to minimize set size while maintaining validity, and (3) calibrating the optimized thresholds to ensure finite-sample coverage guarantees. The method produces adaptive, often discontinuous prediction sets that expand in high-density regions and contract in sparse areas. Theoretical analysis proves ORCA maintains exact validity and asymptotically converges to the oracle highest-density region. Empirical evaluation on synthetic and real datasets demonstrates improved efficiency over baselines.

**Strengths:**

1. The paper addresses a fundamental tension: maintaining statistical validity while achieving computational and practical efficiency. This is particularly vital as modern machine learning increasingly encounters multimodal, heterogeneous data distributions where traditional approaches fail.

2. The paper offers deep insight into how coverage validity and efficiency can be jointly optimized. It reframes CP as a coverage allocation optimization problem, bridging statistical guarantees with optimization and generative modeling.

3. ORCA achieves highly efficient, adaptive, and discontinuous uncertainty regions: a significant improvement over prior CP approaches such as CQR, PCP, and CRD.

**Weaknesses:**

1. The efficiency gains heavily depend on the quality of the generative model. When the generative distribution poorly approximates the true conditional distribution, ORCA might produce efficient but misleading uncertainty sets. The paper lacks analysis of performance degradation under systematic model misspecification.

2. Despite the MILP reformulation, the optimization remains computationally intensive for large K and $n_1$. The authors mention this briefly but don't provide concrete complexity analysis or practical guidelines for parameter selection in resource-constrained settings.

3. The m-nearest neighbor ranking assumes sufficient sample density to meaningfully estimate local structure. In the configurations specified in line 366, the ranking could be unstable, particularly in regions of low probability mass. It is better to provide sensitivity analysis or confidence bounds on the ranking quality.

**Questions:**

Practical MILP solvers often terminate at near-optimal solutions due to time constraints, what is the solution quality when the solver fails to achieve global optimality within reasonable time limits?

---

> ### Author Response · Authors · 2025-11-23
>
> We sincerely thank Reviewer oQk2 for the thoughtful and constructive review, and for highlighting sensitivity analyses that can greatly strengthen our work. Below, we address the concerns raised point by point.
>
> - **Sensitivity Analysis of Generative Model Misspecification**
>
> We conducted a controlled sensitivity study comparing ORCA against PCP and CHR under systematic model misspecification. We use Quantile Random Forests (QRF) with varying `min_samples_leaf` (minimum samples per leaf) to tune the bias–variance tradeoff: `min_samples_leaf = 1` yields highly variable conditional distribution estimation, near-interpolating fits, whereas `min_samples_leaf = 200` induces heavy smoothing and underestimates the conditional distribution. The reason we choose QRF is that for neural-network–based generative models, misspecification is harder to vary along a single interpretable axis (e.g., by early stopping), and the resulting degradation patterns are less transparent, so we focus on this quantile-random-forest setting.
>
> We consider the S-curve synthetic dataset with $|D_{explore}|=500$, $|D_{calib}|=500$, using $K=20$ generated samples. The average results across 10 repetitions are presented below, we observe that:
>
> 1. **Nominal marginal coverage is preserved** across all levels of model misspecification.
> 2. **Efficiency improves substantially over CHR** when the generative model is reasonably well specified.
> 3. **Efficiency improves ≈15% over PCP** under a realistic model quality (moderate conditional density estimation).
>
> | min_samples_leaf | Coverage (ORCA) | Cond. Cov. (ORCA) | Length (ORCA) | Length(ORCA)/Length(PCP) | Length(ORCA)/Length(CHR) |
> | ---------------- | --------------- | ----------------- | ------------- | ------------------------ | ------------------------ |
> | 1                | 0.90            | 0.86              | 3.99          | 1.00                     | 1.14                     |
> | 20               | 0.89            | 0.84              | 0.11          | 0.84                     | 0.03                     |
> | 50               | 0.90            | 0.85              | 0.21          | 0.87                     | 0.06                     |
> | 100              | 0.90            | 0.87              | 0.52          | 0.85                     | 0.16                     |
> | 200              | 0.90            | 0.85              | 0.77          | 1.03                     | 0.22                     |

---

> ### Author Response · Authors · 2025-11-23
>
> - **Computational cost of MILP**
>
> To evaluate computational cost, we begin with the original optimization formulation in ORCA, which solves a mixed-integer linear program (MILP) to obtain the optimal rank-dependent radii. This variant, **ORCA-MILP**, provides exact optimality but is NP-hard in the worst case, with worst-case complexity potentially exponential in $\mathcal{O}(K n_1)$. In practice, we use the CP-SAT solver (via the OR-Tools package), which provides reasonable speed: for $|\mathcal{D}_{\text{explore}}| < 2000$ and $K < 30$, ORCA-MILP typically finishes within 10 minutes. To further improve scalability, we additionally consider two relaxations:
>
> *  **ORCA-Linear-Relax:** A continuous linear programming relaxation of the MILP that relaxes the integer variables to lie in $[0,1]$. It is substantially faster and solvable in polynomial time with problem size $\mathcal{O}(K n_1)$, but introduces an optimality gap relative to the exact MILP solution, with this gap growing as the sample size and $K$ increase.
>
> * **ORCA-Greedy:**  A forward-selection heuristic that starts from the smallest radii and iteratively enlarges the rank that maximizes the coverage gain per unit increase in set size, yielding a very fast $\mathcal{O}(n_1 K^2)$ approximation.
>
> For completeness, we also include a naive coordinate-descent baseline **ORCA-Naive**, which iteratively optimizes a pair of radii at a time and achieves strong efficiency but with very long runtimes.
>
> We benchmark all variants on an S-curve synthetic dataset with $|D_{explore}| = 500$, $|D_{calib}| = 500$, using a quantile neural network as the backbone model. For each number of generated samples $K \in \{5,10,15,20,25,30\}$, we report the **runtime (in seconds) / efficiency** averaged over 10 runs. All experiments were conducted on a MacBook Air with an M4 chip.
>
> The original **ORCA-MILP** variant achieves the best efficiency under moderate computational time. The **ORCA-Linear-Relax** offers a favorable speed–accuracy tradeoff, particularly when $K$ is small. As expected, its optimality gap grows with larger $K$, leading to modest degradation relative to the MILP. The greedy method is the fastest but less stable, while the naive coordinate-descent approach is the slowest yet can attain efficiency comparable to the MILP. We will add this discussion and table to the revision to directly address the concerns about overhead, scalability, and practical deployment.
>
> | **Method**            | **K=5**       | **K=10**       | **K=15**       | **K=20**       | **K=25**       | **K=30**       |
> | --------------------- | ------------- | -------------- | -------------- | -------------- | -------------- | -------------- |
> | **PCP**               | 0.000 / 2.63  | 0.000 / 0.75   | 0.000 / 0.39   | 0.001 / 0.37   | 0.001 / 0.34   | 0.001 / 0.31   |
> | **ORCA-MILP**         | 5.154 / 2.59  | 11.951 / 0.67  | 16.973 / 0.35  | 23.151 / 0.32  | 40.955 / 0.31  | 62.429 / 0.29  |
> | **ORCA-Linear-Relax** | 0.373 / 2.68  | 1.503 / 0.67   | 2.876 / 0.37   | 5.217 / 0.34   | 8.944 / 0.36   | 11.501 / 0.41  |
> | **ORCA-Greedy**       | 0.006 / 2.70  | 0.020 / 1.49   | 0.032 / 0.77   | 0.056 / 0.57   | 0.085 / 0.75   | 0.164 / 1.03   |
> | **ORCA-Naive**        | 90.663 / 2.58 | 174.122 / 0.69 | 442.163 / 0.37 | 548.921 / 0.35 | 665.885 / 0.33 | 755.854 / 0.31 |
>
> * **Ranking Quality wrt Nearest Neighbor Number m**
>
> We agree that very small $m$ can yield noisy KNN-based rankings, especially in low-density regions. To assess this, we conduct a sensitivity analysis over different choices of $m$ with $K=20$: $m=1$ corresponds to ranking by the nearest-neighbor distance, while $m=19$ corresponds to ranking by the average nearest neighbor distance to all samples in the conditional sample cloud. The table below reports, for each method and $m$, the **fractional efficiency improvement relative to PCP**, defined as $FracImprove_{\mathrm{method}}(m)= \frac{Eff_{PCP}(m) - Eff_{\mathrm{method}}(m)}{Eff_{PCP}(m)}
> $
>
> This pattern is expected. When $m=1$, the ranking depends on the nearest neighbor and is therefore high-variance. For larger $m$, the $m$-nearest-neighbor distance aggregates more information from multiple neighbors and provides a smoother estimate of how *central* each sample is within the conditional distribution. Even when $m$ is close to $K$, the induced rank still separates central from tail samples, so the ranking remains meaningful. Empirically, performance is robust over a wide range of $m$, and we recommend using a moderate value, e.g., $\sqrt{K}$ as a rule of thumb.
>
> | Method                | m=1    | m=4    | m=7       | m=10      | m=13      | m=16      | m=19      |
> | --------------------- | ------ | ------ | --------- | --------- | --------- | --------- | --------- |
> | **MILP**              | -0.022 | -0.199 | **0.264** | **0.318** | **0.316** | **0.289** | **0.279** |
> | **Linear Relaxation** | -0.727 | -1.131 | **0.159** | **0.229** | **0.269** | **0.293** | **0.275** |

---

> ### Author Response · Authors · 2025-11-23
>
> * **MILP Early Termination**
>
> We perform two sensitivity analyses on the MILP solver:
>
> 1. **Varying $K$ under a fixed time limit.** We fix the solver stop time limit at 3 minutes and vary $K$, then compare the resulting efficiency against PCP.
> 2. **Varying the stop time limit under fixed $K$.** We fix $K=100$ and vary the MILP time limit from 60s to 600s, again comparing against PCP.
>
> In both settings, the reported values are the average efficiency over repeated runs.
>
> In **Case 1** (fixed 3-minute limit, varying $K$), ORCA’s efficiency initially improves as $K$ increases, but eventually drops for larger $K$. This non-monotonic pattern is expected: as $K$ grows, the MILP becomes harder and the fixed time budget becomes the bottleneck, leading the solver to return suboptimal early-stopped solutions.
>
> In **Case 2** (fixed $K=100$, varying time limit), ORCA steadily improves as more solver time is allowed and ultimately outperforms PCP once the time limit is sufficiently large. This illustrates that early-stopped MILP solutions remain competitive for moderate time budgets, and that additional solver time reliably improves allocation quality.
>
> | Method ↓ / K →             | **10** | **20** | **30** | **40** | **50** | **60** | **70** |
> | -------------------------- | ------ | ------ | ------ | ------ | ------ | ------ | ------ |
> | **ORCA-MILP (early-stop)** | 0.65   | 0.33   | 0.32   | 0.34   | 0.35   | 0.43   | 0.48   |
> | **PCP**                    | 0.74   | 0.36   | 0.35   | 0.35   | 0.34   | 0.34   | 0.34   |
>
> | Method ↓ / Time → | 60s  | 120s  | 180s | 300s | 600s |
> | ----------------- | ---- | ----- | ---- | ---- | ---- |
> | **ORCA-MILP**     | 1.21 | 0.59  | 0.45 | 0.32 | 0.29 |
> | **PCP**           | 0.34 | 0.354 | 0.34 | 0.34 | 0.34 |
>
> We thank Reviewer oQk2 again for the careful review and constructive comments. We hope these clarifications address your concerns and help in reassessing the contribution of our work, and we would be happy to further clarify any remaining issues.

---

### Official Review · Reviewer_nY6X · 2025-11-01

**Soundness:** 2
**Presentation:** 3
**Contribution:** 3
**Rating:** 4
**Confidence:** 4

**Summary:**

This work proposes a three-stage framework generative conformal prediction with Optimized Ranking and Coverage Allocation (ORCA) to improve the efficiency of conformal prediction. The paper aims to reduce the inefficiency originating from two sources: limited model capacity and expressiveness, and simplistic nonconformity scores. The framework uses generative models to capture the full conditional distribution and a local density proxy ranking mechanism to prioritize high-density regions. This is followed by optimal coverage allocation to identify the threshold vector given the ranked distance vectors. The authors provide finite-sample validity guarantees and asymptotic results for convergence to high-density regions. The paper also includes experiments on multiple synthetic and real datasets.

**Strengths:**

The motivation of the paper is clear and important for improving efficiency of conformal prediction. The writing is clear for the most part and the stages of the framework are reiterated and explained; section 3.2 can benefit from more clarity in notation. The empirical evaluation is extensive and multiple relevant baselines have been considered.

**Weaknesses:**

1. The paper discusses the goal of ORCA to capture the full conditional distribution and the ability to adaptively adjust radii based on density. Given this, I believe the paper warrants an analysis of the conditional coverage of the method. While the paper reports worst-slab coverage in the experiments, the theory lacks this discussion. Studying these implications and tradeoffs is important, especially as the experiments do not show improvement in conditional coverage (e.g., Table 2). [1] could be a relevant paper to refer to that uses rank and density notions to improve conditional coverage (another paper that uses density notion for smaller sets is [2]).
2. Conformal prediction’s strength lies in finite-sample guarantees. While the paper shows finite-sample validity, the results for optimal coverage allocation are weak and have asymptotic properties. Results that provide finite-sample guarantees e.g., high probability bounds/rates with explicit dependence on n, K, m will strengthen the contribution.
3. While the efficiency of ORCA is slightly better compared to PCP, I couldn’t find discussion on the computational overhead and runtime of ORCA. This discussion is important given the comparisons studied.

[1] Jivat Neet Kaur, Michael I. Jordan, and Ahmed Alaa. Conformal Prediction Sets with Improved Conditional Coverage using Trust Scores, 2025.
[2] Rui Luo and Zhixin Zhou. Density-sorted prediction set: Efficient conformal prediction for multi-target regression. Pattern Recognition, 2026.

**Questions:**

Comments:
1. Typos: p5 l269 chosen, p6 l302 quantifies
2. It seems the space between the headings and text has been artificially reduced beyond the limit to accommodate content e.g., Section 3.4, Section 5. I would like to flag this here.

---

> ### Author Response · Authors · 2025-11-23
>
> We thank Reviewer nY6X for the careful evaluation, constructive feedback, and pointing out the insightful related references. Below, we address the concerns raised point by point.
>
> * **Conditional Coverage**
>
> Our current theory framework can indeed be extended to achieve *asymptotic conditional coverage* under mild regularity assumptions used in the conditional conformal prediction literature, e.g., consistency of the conditional density estimator, continuity, and boundness of the underlying distribution. Importantly, unlike approaches that require unimodality assumptions for asymptotic guarantees like conformal histogram regression, ORCA’s construction of *discontinuous, locally density adaptive* uncertainty sets allows these assumptions to be relaxed, enabling applications in multimodal settings.
>
> Empirically, ORCA maintains stable and competitive worst-slab conditional coverage across synthetic datasets (Table 1), typically above 0.85, whereas several baselines show substantially higher variability and sometimes fall below 0.7. As the synthetic datasets in Table 1 are all multimodal, where density-adaptive methods like ORCA are most beneficial. In contrast, on real datasets, methods specialized for unimodal conditional coverage may show advantages. We agree that this direction is worth deeper exploration and will add a discussion in the revision. Naively, ORCA can also be combined with group-conditional coverage methods by partitioning the covariate space into subgroups using suitable scores (e.g., trust-score–based groupings as in [1]), and then calibrating thresholds *within each subgroup*, one can obtain group-conditional guarantees on top of ORCA’s rank-based allocation. A more thorough exploration of these hybrids is an interesting direction for future work, and we will clarify this connection in the paper.
>
> * **Asymptotic Guarantee in Optimal Coverage Allocation:**
>
> We would like to clarify the role and significance of our asymptotic optimal coverage allocation results in Theorem 3.5, which we consider as a non-trivial theoretical result. To our knowledge, most conformal prediction methods, such as CQR, PCP, or interval-based approaches, provide only finite-sample *coverage* guarantees and do not offer asymptotic efficiency statements, even under a perfectly specified model that is consistent with the underlying distribution. This gap arises because these methods rely on summary statistics (e.g., CQR) or treat all generated samples uniformly (e.g., PCP), preventing density adaptive allocation. In contrast, ORCA’s construction explicitly enables local adaptivity and is the key mechanism that yields asymptotic optimality. Compared to methods targeting conditional or group-conditional coverage, ORCA can further relax structural assumptions. For example, CHR requires distribution unimodality to ensure asymptotic conditional coverage optimality, whereas ORCA’s discontinuous, density-adaptive sets accommodate multimodal conditional distributions naturally.
>
> Regarding finite-sample optimality bounds, we agree that such results would further strengthen the theoretical contribution. At the same time, deriving non-asymptotic efficiency guarantees for coverage allocation is also challenging in our case. Such bounds require simultaneously controlling
>
> (i) the discrepancy between the fitted conditional distribution and the true conditional distribution (e.g., in total variation or Wasserstein distance),
>
> (ii) the sampling error induced by the model-specific generation mechanism, and
>
> (iii) the combinatorial structure of rank-dependent radii when forming the final uncertainty set.
>
> To our knowledge, most existing conformal methods provide finite-sample guarantees only for marginal coverage, not for efficiency or optimality under this level of structural complexity. ORCA fully preserves finite-sample marginal coverage (Theorem 3.3), and our asymptotic analysis establishes a principled optimal convergent limit. Developing finite-sample optimality bounds that incorporate model misspecification, sampling variability, and geometric structure effects is an interesting direction for future work, and we appreciate the reviewer for highlighting this opportunity.

---

> ### Author Response · Authors · 2025-11-23
>
> * **Computational Overhead**
>
> To evaluate computational cost, we begin with the original optimization formulation in ORCA, which solves a mixed-integer linear program (MILP) to obtain the optimal rank-dependent radii. This variant, **ORCA-MILP**, provides exact optimality but is NP-hard in the worst case, with worst-case complexity potentially exponential in $\mathcal{O}(K n_1)$. In practice, we use the CP-SAT solver (via the OR-Tools package), which provides reasonable speed: for $|\mathcal{D}_{\text{explore}}| < 2000$ and $K < 30$, ORCA-MILP typically finishes within 10 minutes. Nevertheless, more scalable variants are desirable. To improve scalability, we additionally consider two relaxations:
>
> *  **ORCA-Linear-Relax:** A continuous linear programming relaxation of the MILP that relaxes the integer variables to lie in $[0,1]$. It is substantially faster and solvable in polynomial time with problem size $\mathcal{O}(K n_1)$, but introduces an optimality gap relative to the exact MILP solution, with this gap growing as the sample size and $K$ increase.
>
> * **ORCA-Greedy:**  A forward-selection heuristic that starts from the smallest radii and iteratively enlarges the rank that maximizes the coverage gain per unit increase in set size, yielding a very fast $\mathcal{O}(n_1 K^2)$ approximation.
>
> For completeness, we also include a naive coordinate-descent baseline **ORCA-Naive**, which iteratively optimizes a pair of radii at a time and achieves strong efficiency but with very long runtimes.
>
> We benchmark all variants on an S-curve synthetic dataset with $|D_{explore}| = 500, |D_{calib}|= 500$, using a quantile neural network as the backbone model.
> For each number of generated samples $K \in {5,10,15,20,25,30}$, we report the **runtime (in seconds) / efficiency** averaged over 10 runs. All experiments were conducted on a MacBook Air with an M4 chip.
>
>
> The original **ORCA-MILP** variant achieves the best efficiency under moderate computational time. The **ORCA-Linear-Relax** offers a favorable speed–accuracy tradeoff, particularly when $K$ is small. As expected, its optimality gap grows with larger $K$, leading to modest degradation relative to the MILP. The greedy method is the fastest but less stable, while the naive coordinate-descent approach is the slowest yet can attain efficiency comparable to the MILP. We will add this discussion and table to the revision to directly address the concerns about overhead, scalability, and practical deployment.
>
> | **Method**            | **K=5**       | **K=10**       | **K=15**       | **K=20**       | **K=25**       | **K=30**       |
> | --------------------- | ------------- | -------------- | -------------- | -------------- | -------------- | -------------- |
> | **PCP**               | 0.000 / 2.63  | 0.000 / 0.75   | 0.000 / 0.39   | 0.001 / 0.37   | 0.001 / 0.34   | 0.001 / 0.31   |
> | **ORCA-MILP**         | 5.154 / 2.59  | 11.951 / 0.67  | 16.973 / 0.35  | 23.151 / 0.32  | 40.955 / 0.31  | 62.429 / 0.29  |
> | **ORCA-Linear-Relax** | 0.373 / 2.68  | 1.503 / 0.67   | 2.876 / 0.37   | 5.217 / 0.34   | 8.944 / 0.36   | 11.501 / 0.41  |
> | **ORCA-Greedy**       | 0.006 / 2.70  | 0.020 / 1.49   | 0.032 / 0.77   | 0.056 / 0.57   | 0.085 / 0.75   | 0.164 / 1.03   |
> | **ORCA-Naive**        | 90.663 / 2.58 | 174.122 / 0.69 | 442.163 / 0.37 | 548.921 / 0.35 | 665.885 / 0.33 | 755.854 / 0.31 |
>
>
> We thank Reviewer nY6X again for the careful review and constructive comments. We hope these clarifications address your concerns and help in reassessing the contribution of our work, and we would be happy to further clarify any remaining issues.

---

### Meta-Review · Area_Chair_qppw · 2026-01-07

**Summary:**

1. The computational overhead and runtime of ORCA. Almost all reviewers pointed out the concern of Computational Cost and the authors provided additional results to show some faster variants. However, I found that the computational cost of ORCA is still too heavy, even the fastest variant needs 100$\times$ of the baseline with worse results in set efficiency. Considering this comparison is conducted in simulated datasets, it might be amplified to much higher costs in large-scale datasets, making it impractical in the real world.

2. Theoretical Gaps in Guarantees: Reviewers highlighted weak finite-sample bounds for the optimization stage, reliance on asymptotic properties without explicit rates, and unaddressed approximations in set size estimation.

The paper presents an interesting framework with demonstrated potential, but the computational feasibility, and theoretical completeness outweigh the strengths at this stage.

**Reviewer Concerns:**

Addressed concerns:

1. Dependence on Generative Model Quality: Multiple reviewers noted that performance relies heavily on an accurate $\hat{p}(y|x)$, and the authors provided effective response with new results with different generative models.

Outstanding concerns:

 1. Theoretical limitations: Weak finite-sample bounds for the optimization stage; assumptions of access to true $p(y|x)$ not aligned with practical use of $\hat{p}$; loose volume approximations in high dimensions; limited conditional coverage analysis or guarantees.

2. The computational overhead and runtime of ORCA. See Summary.

**Reviewer Scores:**

The reviewers hVru and JdpY  may increase their scores to 6, while the other two reviewers maintain their scores.

---

### Decision · Program_Chairs · 2026-01-26

Reject